# Regulator of G protein signaling 12 enhances osteoclastogenesis by suppressing Nrf2-dependent antioxidant proteins to promote the generation of reactive oxygen species

Andrew Ying Hui Ng[1,2,3†‡], Ziqing Li[1†], Megan M Jones[2], Shuting Yang[1], Chunyi Li[2], Chuanyun Fu[4], Chengjian Tu[3,5], Merry Jo Oursler[6], Jun Qu[3,5], Shuying Yang[1,7*]

[1]Department of Anatomy and Cell Biology, School of Dental Medicine, University of Pennsylvania, Philadelphia, United States; [2]Department of Oral Biology, School of Dental Medicine, University at Buffalo, Buffalo, United States; [3]New York State Center of Excellence in Bioinformatics and Life Sciences, Buffalo, United States; [4]Department of Stomatology, Shandong Provincial Hospital Affiliated to Shandong University, Jinan, China; [5]Department of Pharmaceutical Science, School of Pharmacy and Pharmaceutical Sciences, University at Buffalo, Buffalo, United States; [6]Division of Endocrinology, Metabolism, Nutrition & Diabetes, Mayo Clinic, Rochester, United States; [7]The Penn Center for Musculoskeletal Disorders, School of Medicine, University of Pennsylvania, Philadelphia, United States

*For correspondence:
shuyingy@upenn.edu

†These authors contributed equally to this work

Present address: ‡Department of Anatomy and Cell Biology, University of Pennsylvania, Philadelphia, United States

Competing interests: The authors declare that no competing interests exist.

**Abstract** Regulators of G-protein Signaling are a conserved family of proteins required in various biological processes including cell differentiation. We previously demonstrated that Rgs12 is essential for osteoclast differentiation and its deletion in vivo protected mice against pathological bone loss. To characterize its mechanism in osteoclastogenesis, we selectively deleted Rgs12 in C57BL/6J mice targeting osteoclast precursors using *LyzM*-driven Cre mice or overexpressed Rgs12 in RAW264.7 cells. Rgs12 deletion in vivo led to an osteopetrotic phenotype evidenced by increased trabecular bone, decreased osteoclast number and activity but no change in osteoblast number and bone formation. Rgs12 overexpression increased osteoclast number and size, and bone resorption activity. Proteomics analysis of Rgs12-depleted osteoclasts identified an upregulation of antioxidant enzymes under the transcriptional regulation of Nrf2, the master regulator of oxidative stress. We confirmed an increase of Nrf2 activity and impaired reactive oxygen species production in Rgs12-deficient cells. Conversely, Rgs12 overexpression suppressed Nrf2 through a mechanism dependent on the 26S proteasome, and promoted RANKL-induced phosphorylation of ERK1/2 and NFκB, which was abrogated by antioxidant treatment. Our study therefore identified a novel role of Rgs12 in regulating Nrf2, thereby controlling cellular redox state and osteoclast differentiation.
DOI: https://doi.org/10.7554/eLife.42951.001

## Introduction

Osteoporosis is a pervasive disorder characterized by skeletal fragility and microarchitectural deterioration that predisposes individuals to bone fractures. The disease has a significant global impact,

**eLife digest** Human bodies change with age, and the skeleton is among the parts of the body most visibly affected. This is because bone tissue tends to decrease as the skeleton gets older. For example, people often get shorter as they get older, mostly because they lose bone mass in areas of the skeleton that support posture. Severe bone loss can also lead to osteoporosis, a debilitating condition where bones become brittle and fracture easily.

Human skeletons contain cells, called osteoclasts, which break down bone tissue. Osteoclasts normally cooperate with other cells that add new bone, which helps maintain the balance between 'bone-eating' and 'bone-building' responsible for sculpting a healthy skeleton. This balance is disrupted during old age when the body starts producing too many 'hyperactive' osteoclasts, and bone formation cannot keep up with bone loss.

Reactive oxygen species (ROS) are unstable, potentially toxic molecules that have been linked with diseases of aging. Recent research has shown that low amounts of ROS can also drive the formation of new osteoclasts. Ng, Li et al. therefore wanted to determine how exactly ROS did this – specifically, whether ROS works together with the cell signaling mechanisms involved in bone loss controlled by a gene called *Rgs12*.

Initial experiments, using genetically altered mice, showed that removing *Rgs12* from immature osteoclasts was enough to stop them from maturing. The bones of these mice were also stronger and thicker than usual. In contrast, forcing osteoclasts to produce large amounts of the protein encoded by *Rgs12* heightened their bone-eating ability.

Analysis of the proteins made by cells without *Rgs12* revealed that the cells had turned on the *Nrf2* gene, a molecular 'master switch' that helps produce the enzymes capable of counteracting ROS (termed antioxidants). These cells therefore contained abnormally high amounts of antioxidants and low levels of ROS. However, osteoclasts where the *Rgs12* gene was present were able to generate ROS by switching off the *Nrf2* gene, and were thus able to reach maturity.

These results shed new light on the molecular signals that direct the development and activity of osteoclasts. In the future, a better understanding of these mechanisms could help us prevent them going wrong during aging, or even lead to better therapies for osteoporosis and other skeletal disorders.

DOI: https://doi.org/10.7554/eLife.42951.002

affecting an estimated 200 million people worldwide and exerts a heavy economic burden. Moreover, disease prevalence is projected to rise by approximately 50% within the next ten years (*Amin et al., 2014*). Therefore, understanding the pathogenesis of osteoporosis is an urgent matter to develop better treatments for this debilitating disease.

Bone remodeling is carried out by the coordinated actions of the bone-forming osteoblasts (OBs) and the bone-resorbing osteoclasts (OCs). Disorders of skeletal deficiency such as osteoporosis are typically characterized by enhanced osteoclastic bone resorption relative to bone formation, thereby resulting in net bone loss. Although significant progress has been made in understanding the pathological role of OCs, the molecular pathways that drive OC differentiation remains an area needing further investigation.

Regulators of G-protein Signaling (RGS) are a family comprised of more than thirty proteins that share the eponymous and functionally conserved RGS domain; these proteins play a classical role in attenuating G protein-coupled receptor signaling through its GTPase-accelerating protein activity to inactivate the $G\alpha$ subunit (*Neubig and Siderovski, 2002*; *Keinan, 2014*). RGS proteins are multifunctional proteins that can contribute to various cellular processes including cell differentiation. Rgs12 is unique in that it is the largest protein in its family. In addition to the RGS domain, Rgs12 contains a PSD-95/Dlg/ZO1 (PDZ) domain, a phosphotyrosine-binding (PTB) domain, a tandem Ras-binding domain (RBD1/2), and a GoLoco interaction motif. The multi-domain architecture of Rgs12 is thought to facilitate its role as a scaffolding protein in complexes in which multiple signaling pathways might converge (*Snow et al., 2002*; *Sambi et al., 2006*; *Snow et al., 1998*; *Willard et al., 2007*; *Schiff et al., 2000*).

Reactive oxygen species (ROS) are produced as a normal byproduct of cellular metabolism (*Callaway and Jiang, 2015*) and forms the basis Denham Harman's free radical theory of aging, which perhaps is the most enduring model of aging. The theory of aging postulates that the gradual accumulation of damage inflicted by ROS eventually manifests as degenerative diseases associated with aging (*Harman, 1956*; *Krause, 2007*). In addition to longevity, ROS have been implicated in the management and prevention of cancers, cardiovascular diseases, macular degeneration, Alzheimer's disease, arthritis, and many other tissues—to which the bone is no exception (*Naka et al., 2008*; *Domazetovic et al., 2017*).

More recent studies have shown that RANKL-induced ROS are indispensable for OC differentiation (*Callaway and Jiang, 2015*; *Lee et al., 2005*; *Kim et al., 2010*; *Bartell et al., 2014*). ROS at high levels induce oxidative stress, which if left unchecked becomes deleterious to cell. At low concentrations, however, ROS have been shown to participate in signaling events in OCs, including the RANKL-dependent activation of mitogen-activated protein kinases (MAPKs), phospholipase C gamma (PLCγ), nuclear factor kappa B (NFκB), and $[Ca^{2+}]$ oscillations; all of which contribute to the activation of nuclear factor of T-cells (NFAT), the master regulator of OC differentiation. Multiple lines of evidence have consistently shown that suppression of ROS by various means could inhibit OC differentiation (*Lee et al., 2005*; *Kim et al., 2010*; *Bartell et al., 2014*). In particular, RANKL-dependent activation of PLCγ, $[Ca^{2+}]$ oscillations, and NFAT were abrogated when OC precursors were treated with the antioxidant N-acetylcysteine (NAC) (*Kim et al., 2010*). Furthermore, our previous studies demonstrated that Rgs12 silencing could inhibit PLCγ activation, $[Ca^{2+}]$ oscillations, and the expression of NFATc1 and its downstream factors (*Yang and Li, 2007*). Hence, these findings led us to hypothesize that Rgs12 may play a role in regulating the cellular redox state, thereby controlling OC differentiation.

## Results

### Targeted deletion of Rgs12 selectively reduced osteoclast formation and increased trabecular bone mass

To assess the role of Rgs12 in OC differentiation and bone remodeling in vivo, we generated a conditional gene knockout mouse model by crossing *Rgs12*^flox/flox^ mice with Lysozyme M-cre (LyzM^Cre^) transgenic mice (Rgs12 cKO). The LyzM promoter-driven Cre expression targets *Rgs12* gene deletion to cells of the myeloid lineage, including monocytes/macrophages (*Abram et al., 2014*; *Clausen et al., 1999*). Micro-CT analysis of the distal femurs obtained from *Rgs12*^flox/flox^ and *Rgs12*^+/+^;*LyzM*^Cre^ showed no statistically difference in bone histomorphometry (*Figure 1—figure supplement 1*). *Rgs12*^flox/flox^ mice were used as controls. The Cre-lox-mediated deletion of the *Rgs12* gene was confirmed by PCR amplification of spleen genomic DNA (*Figure 1A*) and qPCR to measure *Rgs12* transcripts in isolated bone marrow macrophages (BMMs) (*Figure 1B*), thereby confirming our mouse Rgs12 cKO model.

Rgs12 cKO mice exhibited increased trabecular bone mass, evident in the hematoxylin and eosin (H and E)-stained sections of the proximal tibia (*Figure 1C*) and 3D micro-computed tomography (micro-CT) visualization of the femoral trabecular bone morphology and microarchitecture (*Figure 1D*). Quantitative micro-CT measurements further demonstrated statistically significant increases in bone volume (VOX-BV/TV), accompanied by increases in both trabecular number (Tb.N) and thickness (Tb.Th), and reduced trabecular separation (Tb.Sp) (*Figure 1E–I*). Therefore, the targeted deletion of Rgs12 in mice resulted in increased bone mass.

To determine whether the high bone mass was a consequence of decreased osteoclast numbers or increased osteoblast numbers in vivo, we quantified the cell numbers from distal femurs stained for tartrate-resistant acid phosphatase (TRAP) (*Figure 1J*). The histological assessment clearly demonstrates no difference in osteoblast numbers between Rgs12 cKO and control bone tissues (*Figure 1K*) whereas the number of TRAP^+^ multinucleated cells were markedly reduced in Rgs12-deficient samples (*Figure 1L and M*). Furthermore, a dynamic histomorphometric analysis by double calcein labeling to measure bone growth over time shows that the rate of bone formation was comparable between Rgs12 cKO and control mice (*Figure 1N–P*). Consequently, the results collectively support the specific role of Rgs12 in OCs and emphasizes the gene's importance in bone remodeling.

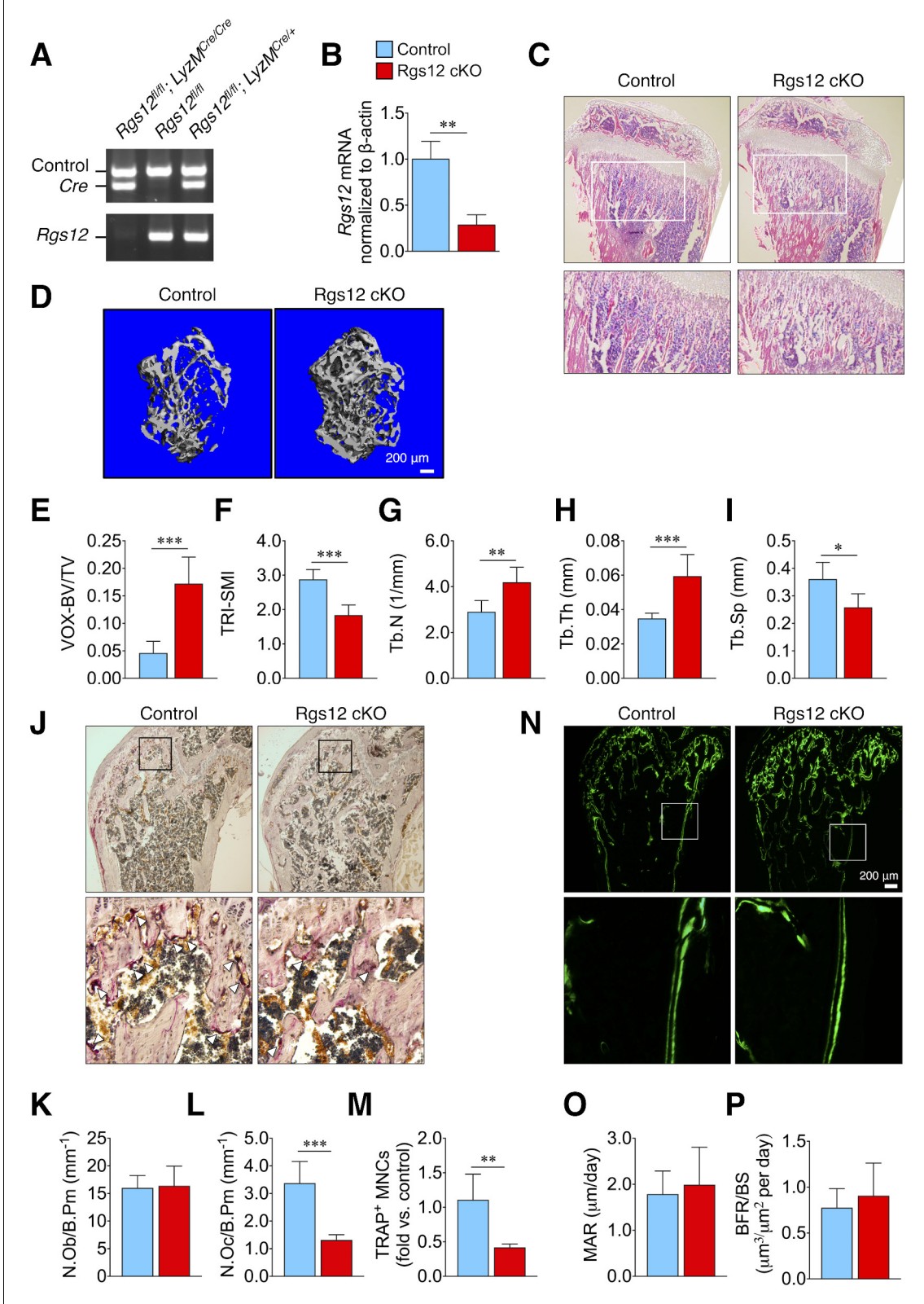

**Figure 1.** Rgs12-deficient mice exhibit increased trabecular bone mass attributed to impaired osteoclastogenesis. (**A**) PCR of splenic genomic DNA amplifying the deletion allele in Rgs12 cKO and control mice. (**B**) qPCR analysis of Rgs12 mRNA levels normalized to β-actin in BMMs obtained from Rgs12 cKO and control mice. Histological assessment of bone morphology and microarchitecture of Rgs12 cKO and control mice include: (**C**) H and E staining of proximal tibiae (N = 4), (**D**) 3D micro-computed tomography (micro-CT) imaging and (**E–I**) quantitative measurement of femoral trabecular

*Figure 1 continued on next page*

*Figure 1 continued*

bone (N$_{Control}$ = 11, N$_{Rgs12cKO}$=7), (J) TRAP staining and quantitation of (K) OBs and (L–M) OCs in distal femurs (N = 5), and (N) dynamic histomorphometry analysis by double-calcein labeling and (O–P) quantitative measurements of bone formation in distal femurs (N = 5). All results are means ± SD (*p<0.05, **p<0.01, ***p<0.001). *VOX-BV/TV, bone volume to tissue volume (voxel count); TRI-SMI, structure model index; Tb.Th, trabecular thickness; Tb.N, trabecular number; Tb.Sp, trabecular separation; N.Ob/B.Pm, osteoblast number per bone perimeter; N.Oc/B.Pm, osteoclast number per bone perimeter; TRAP, tartrate-resistant acid phosphatase; MNC, multinucleated cell; MAR, mineral apposition rate; BFR/BS, bone formation rate per bone surface.*

DOI: https://doi.org/10.7554/eLife.42951.003

The following source data and figure supplement are available for figure 1:

**Source data 1.** Excel sheet contains the numerical data and summary statistics representing the micro-CT data in *Figure 1E–I*.
DOI: https://doi.org/10.7554/eLife.42951.005

**Figure supplement 1.** Bone histomorphometry is not significantly different between *Rgs12+/+;LyzM$^{Cre}$* (N = 5) and *Rgs12$^{flox/flox}$* mice (N = 11), but significantly different between *Rgs12+/+;LyzM$^{Cre}$* (N = 5) and *Rgs12$^{flox/flox}$;LyzM$^{Cre}$* mice (N = 7).
DOI: https://doi.org/10.7554/eLife.42951.004

## Rgs12 promotes osteoclast formation and bone resorptive activity

To further evaluate the role of Rgs12 in osteoclastogenesis, OC precursors isolated from wild-type mice (*Rgs12+/+*) were differentiated using macrophage colony-stimulating factor (M-CSF) and receptor activator of nuclear factor κB ligand (RANKL), the two cytokines necessary and sufficient to induce osteoclast formation. Rgs12 protein and transcript levels were dramatically upregulated upon stimulation by the differentiation factors, and seem to consistently increase into OC maturity at day 5 (*Figure 2A–B*). A comparison of osteoclastogenic potential between precursors derived from Rgs12 cKO and control mice show that while control BMMs differentiated into large, TRAP+ multinucleated OCs, Rgs12-deficient precursor cells showed a reduction in the number of OCs containing 6–9 nuclei and 10+ nuclei, which were also visibly smaller (*Figure 2C–D*). We also probed into the overall bone resorptive activity and found that Rgs12-deficient OCs have significantly reduced ability to resorb calcium phosphate surfaces (*Figure 2E–F*). Complementing our Rgs12 knockout model, we generated an Rgs12 overexpression OC model in which the transformed murine macrophage-like RAW264.7 cells were stably-transfected with a vector carrying a recombinant N-terminus FLAG-tagged Rgs12 gene (Flag-Rgs12). Rgs12 overexpression in RAW264.7 cells was confirmed by western blotting (*Figure 2G*). Using this cell model, we next determined whether Rgs12 overexpression could promote OC formation. Contrasting our findings in Rgs12 cKO primary cells, we found that the overexpression of Rgs12 in RAW264.7 cells led to an increased number of OCs with 10+ nuclei (*Figure 2H–I*). We also observed significantly decreased numbers of smaller OCs containing 3–5 and 6–9 nuclei in Rgs12 overexpressing cells, presumably because most of the smaller OCs have fused to form large OCs containing 10+ nuclei. A previous study investigating the relationship between OC size and state of resorptive activity found that a greater proportion of large OCs were active whereas non-resorbing OCs were on average smaller (*Lees et al., 2001*). In our study, the quantification of the mean areas of OCs with 10+ nuclei revealed that Rgs12-overexpressing OCs were significantly larger as compared to empty vector-transfected controls (*Figure 2J*). To determine whether increased OC size translated to increased bone resorptive activity in our study, RAW264.7 cells were similarly cultured on calcium phosphate surfaces (*Figure 2K–L*). Consistent with our overall findings, ectopic overexpression of Rgs12 in OCs potently increased bone resorption activity. Our findings therefore demonstrate the importance of Rgs12 in promoting OC formation and activity, which is consistent with the osteopetrotic phenotype observed in the Rgs12-deficient mouse model.

## Rgs12-deficient osteoclast precursors show an increased expression of Nrf2-dependent antioxidant proteins

To uncover the role of Rgs12 in OC differentiation, we employed the *IonStar* liquid chromatography tandem mass spectrometry (LC-MS/MS)-based quantitative proteomics strategy (*Shen et al., 2018*) to profile the temporal dynamics in the global protein levels in Rgs12 cKO and control BMMs at 0, 1, 3, and 5 days of OC differentiation (*Figure 3*). Proteomics analysis identified 3714 quantifiable proteins that are present in all samples (no missing data), using a highly stringent identification criteria of ≥2 peptides per protein and 1% false discovery rate (*Figure 3A*). Within this dataset, we identified 83 and 61 unique proteins that were significantly up- and downregulated, respectively, in

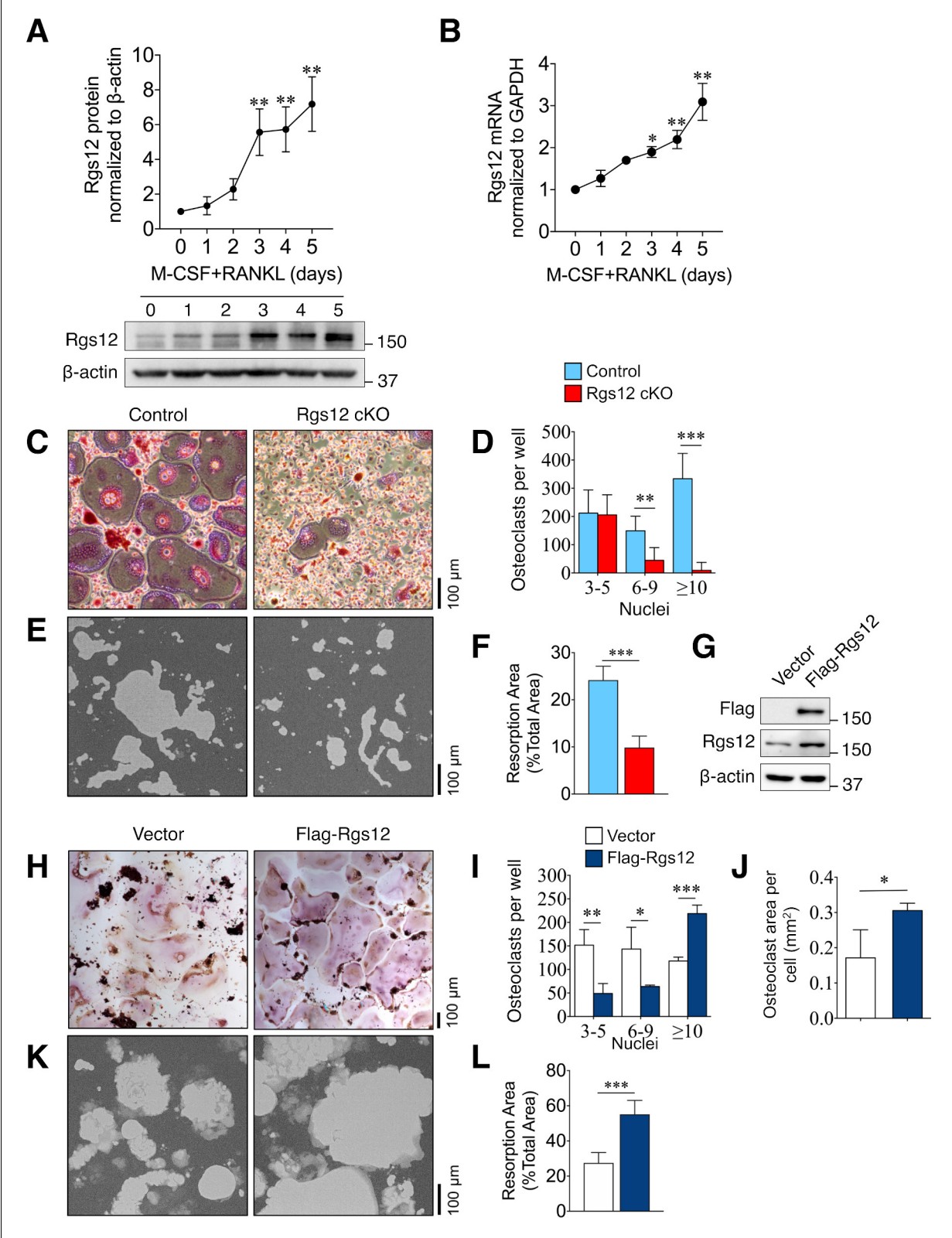

**Figure 2.** Rgs12 is essential for osteoclast differentiation and bone resorption. (**A**) Rgs12 protein and (**B**) mRNA expression in wild-type BMMs stimulated with M-CSF and RANKL for the indicated times. (**C**) TRAP-stained osteoclasts differentiated from BMMs isolated from Rgs12 cKO and control mice and the (**D**) number of TRAP-positive and multinucleated (≥3 nuclei) OCs were counted (N = 4). (**E–F**) Bone resorption activity of OCs derived from Rgs12 cKO and control BMMs cultured on calcium phosphate-coated plastic (N = 5). The light-colored areas correspond to areas

*Figure 2 continued on next page*

*Figure 2 continued*

resorbed by OCs was quantified and presented as values relative to the total area measured. (G) Immunoblot to verify Rgs12 overexpression in RAW264.7 cells transfected with a vector carrying a recombinant N-terminus FLAG-tagged Rgs12 gene (Flag-Rgs12). RAW264.7 cells transfected with the empty vector was used as a negative control. (H) TRAP-stained osteoclasts derived from RAW264.7 cells transfected with an empty vector or Flag-Rgs12 and the (I) number of TRAP-positive and multinucleated (≥3 nuclei) osteoclasts from vector- and Flag-Rgs12-transfected RAW264.7 cells (N = 3). (J) OC size was estimated by quantifying the surface area of OCs containing 10+ nuclei normalized to the number of OCs with 10+ nuclei (N = 3). (K–L) Bone resorption activity of OCs derived from RAW264.7 cells transfected with empty vector or Flag-Rgs12 (N = 5). All results are means ± SD. Student's *t* test was used in all cases except for *Figure 1A and B* wherein one-way ANOVA was used (*p<0.05, **p<0.01, ***p<0.001). TRAP, tartrate-resistant acid phosphatase.

DOI: https://doi.org/10.7554/eLife.42951.006

The following figure supplement is available for figure 2:

**Figure supplement 1.** Complete western blots used for *Figure 2C*.

DOI: https://doi.org/10.7554/eLife.42951.007

Rgs12 cKO OCs relative to control. Proteins were considered significantly altered if they exceeded the empirically-determined thresholds set at p<0.05 and>0.3 $\log_2$-transformed ratio (*Figure 3B*). Most of the protein expression changes in Rgs12-deficient cells were captured at 3 and 5 days of OC differentiation (*Figure 3A*). Interestingly, the proteomic disturbances as a result of Rgs12 deletion closely coincided with the pattern of endogenous Rgs12 protein expression during OC differentiation (*Figure 2A*). To determine the biological significance of these altered proteins, we performed gene ontology analysis to identify the canonical pathways involved (*Figure 3C*). Classically processes related to OC differentiation (e.g. 'NFAT Signaling', 'RANK Signaling in OCs', and 'Role of OCs in Rheumatoid Arthritis') were enriched at 3 and 5 days of OC differentiation. Closer inspection showed that OC marker proteins including metalloproteinase-9 (Mmp9), TRAP, ATPase H$^+$ transporting V0 subunit D2 (Atp6v0d2), and integrin β3 (Itgb3) were significantly downregulated in Rgs12 cKO OCs (*Figure 3D*). Additionally, the analysis revealed several biological functions related to ROS homeostasis that were impacted by Rgs12 deletion (e.g. 'Production of ROS', 'Superoxide Radical Degradation', and NRF2-mediated Stress Response') (*Figure 3C*). Inspection of the proteins involved in these pathways showed a significant upregulation of numerous Nrf2-dependent antioxidant enzymes responsible attenuating oxidative stress, including: peroxiredoxin 1/4 (Prdx1/4), thioredoxin 1/2 (Trxr1), glutathione reductase (Gshr), and NAD(P)H dehydrogenase quinone 1 (Nqo1) (*Figure 3E*). Upstream regulator (transcription factor) analysis by the Ingenuity Pathway Analysis software function identified that the antioxidant enzymes upregulated in Rgs12 cKO OCs share the common upstream regulator Nrf2, a key transcription factor that regulates cellular redox balance through the expression of protective antioxidant and phase II detoxification proteins. (*Venugopal and Jaiswal, 1996*; *Itoh et al., 1997*). Although the upstream regulator analysis predicted an upregulation of Nrf2 activity, the transcription factor itself was not detected by our proteomics analysis. Proteins of typically low abundance such as cytokines, signal regulatory molecules, and transcription factors tend to be 'crowded out' during MS analysis by more highly abundant proteins such those proteins involved in glycolysis and purine metabolism, protein translation, and cytoskeletal components (*Beck et al., 2011*). Nonetheless, our proteomics-based discovery tool allowed us to generate the hypothesis that Nrf2 is aberrantly activated by Rgs12 deletion, causing excessive clearance of ROS by antioxidant enzymes and in turn disrupting OC differentiation.

## Deletion of Rgs12 elevated Nrf2/Keap1 expression and Nrf2 activity

Based on our proteomics analysis, we hypothesized that Rgs12 is needed to suppress Nrf2 activity and facilitate the formation of ROS, which has been previously shown to play a critical role in OC differentiation (*Lee et al., 2005*; *Kanzaki et al., 2013*). To test this hypothesis, we assessed Nrf2 activity and the expression of Nrf2 and Keap1 in Rgs12 cKO and control precursor cells (*Figure 4*). Western blotting of Nrf2 in day 3 OCs also showed increased levels of Nrf2 in Rgs12 cKO cells (*Figure 4A–B*). Keap1, however, which is known to suppresses Nrf2 activity by facilitating its degradation via the proteasome pathway, was unexpectedly elevated in Rgs12-deficient cells. Furthermore, immunofluorescence staining of Nrf2 demonstrated increased nuclear translocation of the transcription factor in Rgs12 cKO cells (*Figure 4C*, upper panel). The elimination of ROS with N-acetylcysteine (NAC), a precursor to the antioxidant glutathione, was able to completely suppress Nrf2

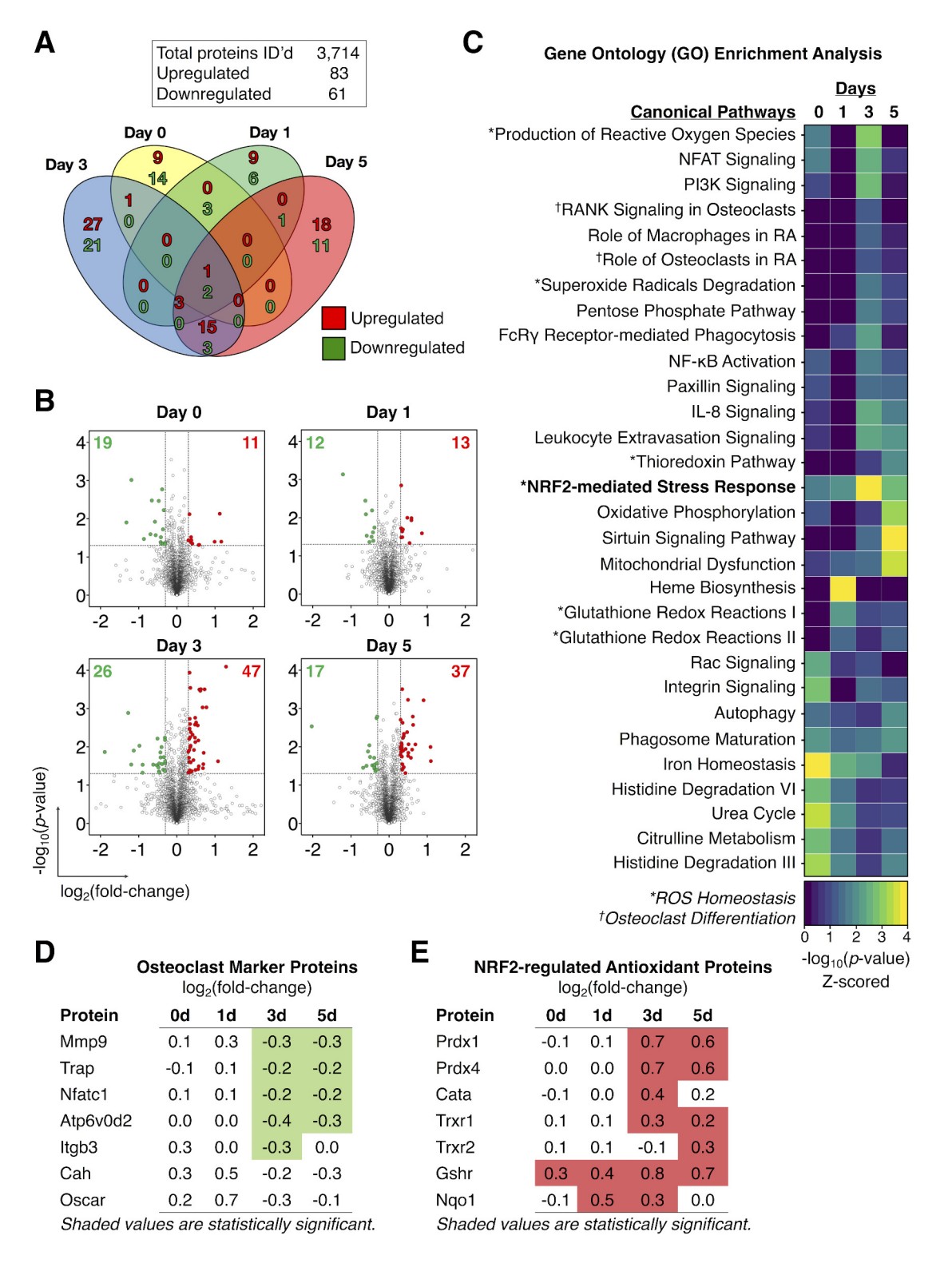

**Figure 3.** Proteomics analysis identified an increased expression of Nrf2-dependent antioxidant proteins in Rgs12-deficient osteoclast precursors. (A) Venn diagram summarizing the distribution of proteins that were significantly altered in Rgs12 cKO BMMs as compared to control at 0, 1, 3, and 5 days of OC differentiation. (B) Volcano plots depicting protein expression changes in Rgs12 cKO BMMs as compared to control cells. Optimized cutoff thresholds for significantly altered proteins was set at 1.3 log2-transformed ratios and p-value<0.05. Data are means ± SD. Student's t test was

*Figure 3 continued on next page*

*Figure 3 continued*

performed to compare Rgs12 cKO and control BMMs at each time point (N = 3). (**C**) Gene ontology (GO) enrichment analysis to identify canonical pathways corresponding to the significantly altered proteins. For visualization purposes, the color intensity in the heat map diagram indicates the significance of GO term enrichment, presented as –log10(P-value). Hierarchical clustering analysis was used to group GO terms based on the p-value of enrichment. (**D–E**) The expression of OC marker proteins and Nrf2-regulated antioxidant proteins in Rgs12 cKO versus control BMMs. Mmp9, metalloproteinase-9; Trap, tartrate-resistant acid phosphatase; Nfatc1, nuclear factor of activated T cells, cytoplasmic 1; Atp6v0d2, ATPase H+ transporting V0 subunit D2; Itgb3, integrin β3; Prdx, peroxiredoxin; Cata, catalase; Trxr, thioredoxin; Gshr, glutathione reductase; Nqo1, NAD(P)H dehydrogenase quinone 1.

DOI: https://doi.org/10.7554/eLife.42951.008
The following source data and figure supplement are available for figure 3:

**Source data 1.** Proteomics data presented in *Figure 3*.
DOI: https://doi.org/10.7554/eLife.42951.011
**Source data 2.** Summary of proteins involved in energy metabolism including glycolysis, TCA cycle, and oxidative phosphorylation.
DOI: https://doi.org/10.7554/eLife.42951.009
**Figure supplement 1.** Proteins in the glycolysis, tricarboxylic acid, and oxidative phosphorylation pathways examined by the Ingenuity Pathway Analysis software.
DOI: https://doi.org/10.7554/eLife.42951.010

nuclear translocation in both Rgs12 cKO and control BMMs (*Figure 4C*, middle panel). Conversely, induction of oxidative stress using the peroxide *tert*-buthylhydroxyperoxide (tBHP) potently induced Nrf2 nuclear translocation (*Figure 4C*, bottom panel). To further test whether elevated Nrf2 activity in Rgs12-deficient OCs could result in reduced intracellular ROS levels, we detected intracellular ROS levels. As expected, the RANKL-dependent ROS induction observed in control cells was suppressed in Rgs12 cKO OCs (*Figure 4D*). These findings demonstrate an abnormal upregulation of Nrf2 activity and expression in Rgs12-deficient cells, indicating that Rgs12 may be required to suppress Nrf2 to facilitate osteoclastogenesis.

## Rgs12-mediated suppression of Nrf2 activity is dependent on the proteasome degradation pathway

Under basal conditions (i.e. absence of cellular stress), Nrf2 remains inactive through its interaction with Keap1, which causes its continual ubiquitination and degradation via the proteasome pathway (*Stewart et al., 2003*; *Zhang and Hannink, 2003*). A variety of stress conditions can induce conformational changes in Keap1, thereby releasing Nrf2 from the ubiquitin-proteasome pathway, allowing it to accumulate and translocate into the nucleus (*Itoh et al., 2003*; *Kensler et al., 2007*). To better understand the mechanism by which Rgs12 suppresses Nrf2 activity, we therefore first determined whether the ability of Rgs12 to suppress Nrf2 activity relies on this canonical mechanism (*Figure 5A*). Given that Rgs12 deletion resulted in elevated Nrf2 expression and nuclear translocation, we first determined whether Rgs12 overexpression could exert an opposite effect (*Figure 5A–B*). We measured Nrf2 protein levels in RAW264.7 cells stably transfected with the Rgs12-His or empty vector and found no difference when cells are at their un-induced, basal state. Stimulation of RAW264.7 cells with *tert*-buthylhydroquinone (tBHQ), which is known to directly bind Keap1 and attenuate its inhibitory effect on Nrf2 (*Abiko et al., 2011*), caused a robust induction of Nrf2 protein levels in a dose-dependent manner (*Figure 5A–B*). More importantly, RAW264.7 cells overexpressing Rgs12 showed a significant reduction of Nrf2 protein levels that resulted from Keap1 inhibition compared to those in the control cells. Moreover, the ability of Rgs12 to facilitate Nrf2 degradation despite the inhibition of Keap1 suggests that Rgs12 functions downstream of Keap1, either by controlling the ubiquitination or proteasomal degradation of Nrf2.

Given the possibility that the reduction of Nrf2 levels in Rgs12 overexpression cells may be a result of increased Nrf2 degradation, we further tested whether inhibiting the proteasome, a step downstream of Keap1, could attenuate the ability of Rgs12 to facilitate Nrf2 degradation (*Figure 5D and E*). Similar to tBHQ, preventing Nrf2 degradation using the proteasome inhibitor MG-132 caused Nrf2 protein to substantially accumulate (*Figure 5D*, left panel). Interestingly, when Nrf2 protein levels were artificially induced, we observed the presence of a lower molecular weight band, which could correspond to a different post-translational modification state (e.g. unphosphorylated or non-ubiquitinated). Furthermore, we did not observe any changes in Keap1 protein levels. In the

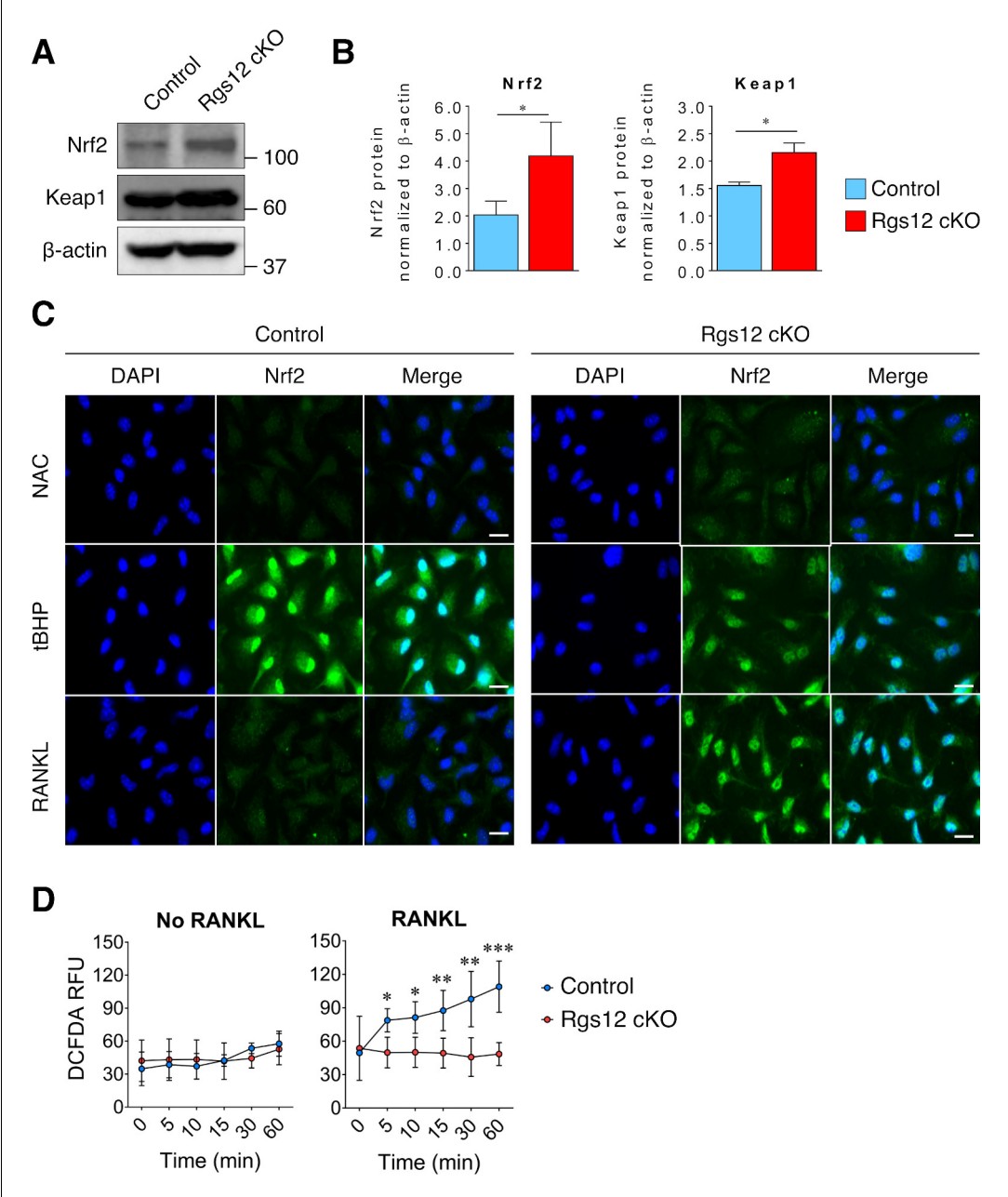

**Figure 4.** Increased Nrf2 activation and expression of antioxidant proteins in Rgs12-deficient osteoclast precursors. (**A–B**) Immunoblot of Nrf2 and Keap1 protein levels in Rgs12 cKO and control BMMs treated with RANKL for 72 hr. Densitometry analysis was performed on bands and normalized to β-actin (N = 3, *p<0.05). (**C**) Nrf2 immunofluorescence staining in Rgs12 cKO and control BMMs differentiated with M-CSF and RANKL for 72 hr. As a negative control for Nrf2 nuclear translocation, cells were treated with the antioxidant compound NAC (5 mM, 16 hr) to suppress cellular ROS. Conversely, as a positive control for Nrf2 nuclear translocation, cells were treated with the peroxidase tBHP (50 μM, 16 hr) to induce oxidative stress. (**D**) Induction of ROS levels in Rgs12 cKO and control BMMs differentiated for 72 hr, kept in serum-free medium for 6 hr, and stimulated with RANKL for the indicated times. ROS levels were measured using the DCFDA fluorescence method. Data are means ± SD (N = 5, *p<0.05, **p<0.01, ***p<0.001). *DAPI, 4,6-diamidino-2-phenylindole; NAC, N acetylcysteine; tBHP, tert-butylhydroxyperoxide. ROS, reactive oxygen species. DCFDA, 2',7'-dichlorofluorescin diacetate. RFU, relative fluorescence units..*
DOI: https://doi.org/10.7554/eLife.42951.012

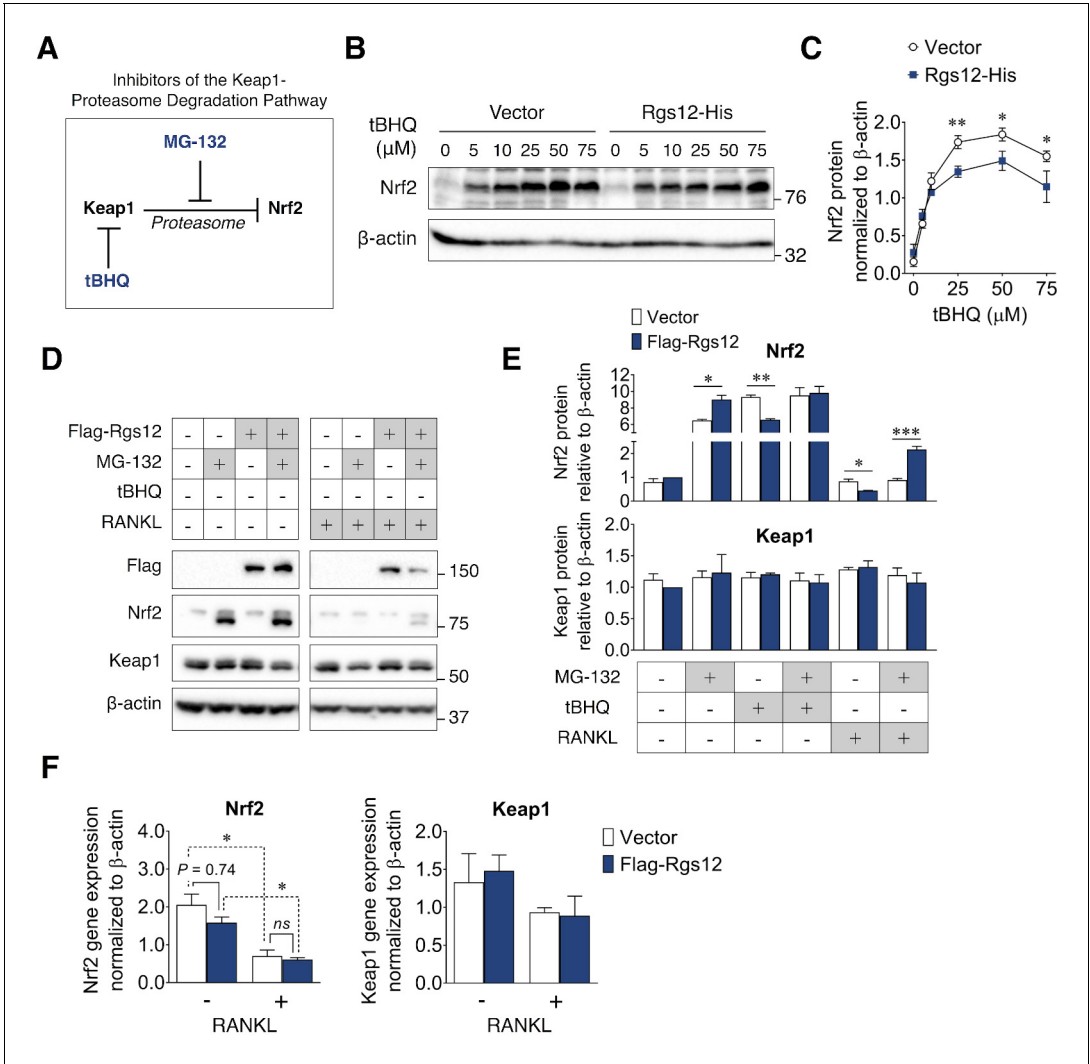

**Figure 5.** Suppression of Nrf2 protein levels by Rgs12 is dependent on the proteasome degradation pathway. (**A**) Diagram summarizing the inhibitors of the Keap1-proteasome axis to modulate Nrf2 protein levels. (**B**) RAW264.7 cells stably-transfected with Rgs12-His or empty vector treated with increasing doses of tBHQ. (**C**) Nrf2 and Keap1 protein levels were quantified by densitometry analysis and normalized to β-actin (N = 3, *p<0.05, **p<0.01). (**D**) Western blot to detect Nrf2 and Keap1 in RAW264.7 cells stably-transfected with empty vector or Flag-Rgs12. RAW264.7 cells were treated with a combination of RANKL (100 ng/mL, 72 hr) and the proteasome inhibitor MG-132 (25 μM, 4 hr). (**E**) Nrf2 and Keap1 protein levels were quantified by densitometry analysis and normalized to β-actin (N = 3, *p<0.05, **p<0.01, ***p<0.001). (**F**) qPCR analysis of Nrf2 and Keap1 transcript levels in RAW264.7 cells transfected with Rgs12-His or empty vector. Data are means ± SD. Two-tailed *t* test was performed (N = 3, *p<0.05). tBHQ, *tert*-butylhydroquinone.

DOI: https://doi.org/10.7554/eLife.42951.013

The following figure supplement is available for figure 5:

**Figure supplement 1.** Complete western blots shown in *Figure 5*.

DOI: https://doi.org/10.7554/eLife.42951.014

previous scenario wherein Rgs12 overexpression could still promote Nrf2 degradation in spite of tBHQ treatment, this was not the case when using MG-132. In fact, inhibiting the proteasome was able to reverse the ability of Rgs12 to promote Nrf2 degradation, indicating the requirement for Rgs12 in the proteasome's function. We subsequently repeated this experiment in RAW264.7 cells differentiated for 3 days with RANKL (*Figure 5D*). Interestingly, RANKL treatment correlated with reduced Nrf2 levels in OCs (*Figure 5D*, right panel). Our observation corroborates with previous findings documenting the suppressive effect of RANKL on Nrf2 expression, which was attributed to reduced transcriptional activity (*Kanzaki et al., 2013*; *Hyeon et al., 2013*). We confirmed this effect

by measuring the *Nrf2* transcript levels by qPCR (Figure F). More importantly, we demonstrate that Rgs12 overexpression could suppress Nrf2 protein levels, but inhibition of the proteasome using MG-132 reversed this effect (*Figure 5D*, right panel). To confirm that Rgs12 inhibits Nrf2 through a post-translational mechanism, we measured Nrf2 transcript levels by qPCR and found no difference between vector control and Rgs12-overexpressing cells (*Figure 5F*). Overall, our data collectively indicate that Rgs12 suppresses Nrf2 activity by facilitating its degradation through the proteasome-dependent pathway.

## Rgs12-mediated activation of osteoclast MAPK ERK1/2 and NFκB signaling is dependent on intracellular ROS

It was previously demonstrated that ROS could act as an intracellular signal mediator OC differentiation, and is required for the RANKL-dependent activation of p38 mitogen-activated protein kinase (MAPK), extracellular signal-regulated kinase (ERK), and NFκB (*Lee et al., 2005*; *Ha et al., 2004*). Given our findings that Rgs12 could suppress the activity of Nrf2 and thereby promoting intracellular ROS, we hypothesized that Rgs12 could promote RANKL-dependent signaling, and that this effect would be abrogated by the addition of an antioxidant (*Figure 6A–B*). As expected, RAW264.7 cells overexpressing Rgs12 demonstrated a more robust activation of ERK1/2 and NFκB phosphorylation but not p38 MAPK. Pretreating Rgs12 overexpressing cells with the antioxidant NAC diminished ERK1/2 activation, and almost completely abrogated NFκB activation. These results support the role of Rgs12 in promoting ROS that is important OC signaling, likely through the suppression of Nrf2 activity.

## Discussion

The importance of ROS in osteoclasts has been underlined by the growing corpus of evidence demonstrating that ROS increased with aging or during inflammation can stimulate bone resorption and exacerbate bone loss (*Callaway and Jiang, 2015*). Targeting ROS in diseases of excess bone resorption such as osteoporosis could therefore represent a novel therapeutic strategy. An important mechanism of cellular ROS clearance relies on the Keap1-Nrf2 pathway, which has been well characterized especially in the context of cancer biology (*Kansanen et al., 2013*); however, the upstream signaling molecules that could regulate the Keap1-Nrf2 axis in OCs remains unknown. Targeting this gap in knowledge, our study uncovered a novel role of the signaling protein Rgs12 in regulating Nrf2, thereby controlling cellular redox state and OC differentiation.

Within this study, we first demonstrated the essential role of Rgs12 in OC differentiation such that myeloid cell-targeted Rgs12 knockout mice exhibited an osteopetrotic phenotype, evidenced by reduced OC numbers but no changes to OB numbers nor bone formation (*Figure 1*). Furthermore, OC precursors isolated from these mice showed reduced in vitro OC differentiation and bone resorptive activity (*Figure 2*). On the contrary, forced overexpression of Rgs12 in RAW264.7 cells significantly promoted OC formation and increased the size of the resultant OCs, which was also associated with increased bone resorptive activity. However, the mechanism by which Rgs12 regulates OC differentiation remains unclear, to which we investigated more deeply using a leading-edge and high-throughput proteomics technique.

Proteomics is a powerful tool that has led to numerous discoveries of proteins and biological processes that drive OC differentiation (*Segeletz and Hoflack, 2016*). Notably, this technique was recently used to map the podosome proteome which helped to advance our understanding of determinants in the macrophage multinucleation process (*Rotival et al., 2015*; *Cervero et al., 2012*), and how metabolism and energy is redirected towards bone resorption in OCs (*An et al., 2014*). Proteomics can therefore provide a broad yet informative overview of the systemic changes in the differentiating OC. To discover the cellular function of Rgs12 in OCs, we employed a robust and high-throughput quantitative proteomics approach to characterize the global protein changes of OCs derived from Rgs12 cKO and cKO BMMs (*Figure 3*). The analysis revealed that most perturbations to the proteome was related to day 3 and 5 of OC differentiation, which coincides with the timing of Rgs12 upregulation in wild-type OCs (*Figure 2A–B*). This overlap suggests that Rgs12 may be more important to later phases of OC differentiation, including cell-cell fusion and bone resorption. The analysis also identified the upregulation of a collection of antioxidant enzymes that are all transcriptionally regulated by the antioxidant response element (ARE) within the promoter region,

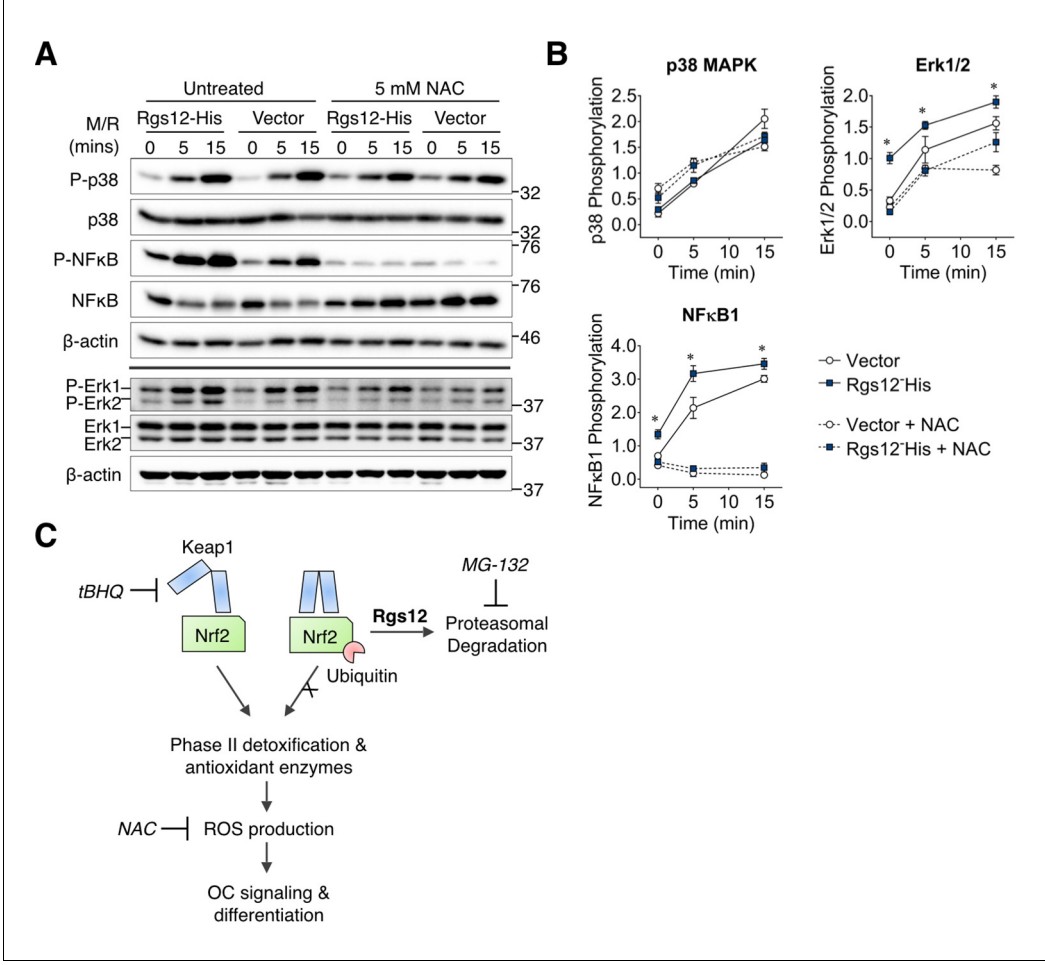

**Figure 6.** Rgs12-dependent activation of ERK1/2 and NFκB was suppressed by antioxidants. (**A**) Western blot detected phosphorylated or total p38, NFκB, and Erk1/2 in transfected RAW264.7 cells induced with RANKL (200 ng/mL) and M-CSF (100 ng/mL) for the indicated times. Cells were pretreated with NAC (5 mM, 4 hr) to suppress intracellular ROS. (**B**) Band density was quantified by ImageJ and phosphorylated and unphosphorylation/total protein levels were normalized to β-actin. Relative phosphorylation is presented as the ratio between the phosphorylated normalized to the nonphosphorylated/total protein. Two-tailed *t* tests were used to compare vector and Rgs12-His groups (N = 3, *p<0.05). (**C**) Model of the role of Rgs12 in suppressing Nrf2 to promote ROS and OC differentiation. M/R, M-CSF and RANKL. NAC, N-acetylcysteine.

DOI: https://doi.org/10.7554/eLife.42951.015

The following figure supplement is available for figure 6:

**Figure supplement 1.** Complete western blots shown in *Figure 6*.

DOI: https://doi.org/10.7554/eLife.42951.016

which is activated by the transcription factor Nrf2 (*Nguyen et al., 2009*). While the upregulation of these genes would lead one to surmise that Nrf2 should be similarly increased, the transcription factor was not detected in the proteomics analysis, which is likely result of technical limitations inherent to current LC-MS capabilities. It was previously reported that transcription factor protein levels are notoriously difficult to obtain due to many having relatively low expression levels that are generally below current mass spectrometer detection limits (*Simicevic and Deplancke, 2017*). The quantitation of transcription factors typically entails an upstream enrichment/separation step before LC-MS analysis. Therefore, the absence of Nrf2 in our proteomics analysis is not an unusual happenstance from a technical standpoint. Alternatively, we confirmed our proteomics findings by showing increased Nrf2 protein levels and nuclear translocation activity in Rgs12-deficient osteoclasts (*Figure 4A–C*).

Based on our preliminary evidence, we further investigated the role of Rgs12 in Nrf2 signaling and found that Nrf2 protein levels and nuclear translocation were increased in OC precursors in which Rgs12 was depleted (*Figure 4*). Because our data showed that Rgs12 deficiency upregulated Nrf2, we expected that Keap1 levels should be reduced in order to facilitate the increased Nrf2 activity. On the contrary, Keap1 levels were upregulated in Rgs12-deficient cells, to which we speculate that Rgs12-deficient cells may be overcompensating Keap1 expression in order to rein back the increased Nrf2 activity. Nevertheless, consistent with the upregulation of Nrf2 and its corresponding antioxidant enzymes, the RANKL-dependent induction of ROS was attenuated in Rgs12-deficient cells. Through an orthogonal approach, we demonstrated that Rgs12 overexpression in OC precursors could also enhance RANKL-mediated activation of ERK1/2 and NFκB (*Figure 6*), which are established to be dependent on ROS (*Hyeon et al., 2013*). Furthermore, inhibition of intracellular ROS blocked the effect of Rgs12 overexpression, indicating that Rgs12 promotes RANKL-dependent signaling by facilitating ROS production. Overall, the data collectively demonstrate that Rgs12 promotes osteoclastogenesis by facilitating ROS generation through the suppression of Nrf2 and its target antioxidant genes.

The transcription factor Nrf2 is constitutively expressed but its activity is inhibited through its interaction with Keap1. Under basal conditions, Nrf2 is restricted to the cytoplasm where it is continually depleted through the proteosomal degradation pathway. When bound to Nrf2, Keap1 recruits the cullin 3 (Cul3)-dependent E3 ubiquitin ligase complex, which ubiquitinates and targets Nrf2 for degradation by the 26S proteasome (*Zhang and Hannink, 2003*; *Itoh et al., 2003*; *Nguyen et al., 2005*; *McMahon et al., 2003*). Keap1 protein contains multiple reactive cysteine residues that serve as redox sensors (*Abiko et al., 2011*) that are sensitive to stressor conditions including oxidative stress causes the electrophilic modification of Keap1, inducing conformational changes, causing the protein to dissociate from Nrf2 and thereby allowing the transcription factor to enter the nucleus (*Stewart et al., 2003*). We therefore determined how Rgs12 regulates this well-defined mechanism (*Figure 5*). tBHQ is a selective inhibitor of Keap1 activity by covalently binding the protein's reactive thiols and as a result activate Nrf2 and its downstream proteins in RAW264.7 cells (*Abiko et al., 2011*). tBHQ was also previously shown to inhibit OC differentiation via the upregulation of heme oxygenase-1, a Nrf2-dependent antioxidant enzyme (*Yamaguchi et al., 2014*). In this study, we determined whether Rgs12 could suppress the tBHQ-dependent upregulation of Nrf2 (*Figure 5B*). We reasoned that if Rgs12 relies on a Keap1-dependent mechanism, then the inhibition of Keap1 by tBHQ should prevent the ability of Rgs12 to suppress Nrf2. However, we observed that Rgs12 was still able to suppress Nrf2 despite the pharmacological blockade of Keap1 activity, thereby indicating that Rgs12 functions downstream of Keap1.

Following Keap1-mediated ubiquitination of Nrf2, the targeted protein should be degraded by the proteasome. Again, we reasoned that if Rgs12 is dependent on the proteasomal degradation pathway, then inhibition of this pathway using the proteasome inhibitor MG-132 should prevent Rgs12-mediated suppression of Nrf2. Indeed, we found that Inhibiting the proteasome was able to reverse the Rgs12-mediated degradation of Nrf2, which places Rgs12 in between Keap1 and the proteasome in the Nrf2 degradation pathway (*Figure 5D–E*). Thus, Rgs12 could either regulate the Cul3-dependent E3 ubiquitin ligase complex to facilitate the ubiquitination of Nrf2, or directly control proteasome activity. While Keap1 certainly plays a canonical role in Nrf2 suppression, the evidence herein suggest that Keap1 is not involved in the Rgs12 function.

It is also interesting to note that NFκB activation is also dependent on the proteasomal degradation of inhibitor of κB (IκB), which otherwise sequesters NFκB to the cytoplasm (*Boyce et al., 2015*). If Rgs12 could modulate proteasome activity, it is possible that Rgs12 could directly promote NFκB by facilitating the degradation of IκB. However, the fact that antioxidant treatment to suppress ROS could almost completely block the phosphorylation of NFκB points to an important involvement of ROS, and not just simply the proteasomal degradation of IκB in NFκB activation (*Figure 6*). Nonetheless, this potential crosstalk between the NFκB and Nrf2 pathways will need to be evaluated in future studies.

ROS is an inevitable byproduct of the mitochondrial electron transport chain (oxidative phosphorylation, OXPHOS) during ATP synthesis. Microscopy analyses have noted the high abundance of mitochondria in osteoclasts as early as 1961 (*Gonzales, 1961*), but interest in the metabolic profile that supports OC differentiation has made a resurgence in recent years. As examples, we and others through the proteomic study of OCs have reported that proteins involved in ATP synthesis were

elevated in mature osteoclasts as compared to precursors (*An et al., 2014*; *Ng et al., 2018*). Additionally, it was demonstrated that mature OCs exhibit larger mitochondria, higher levels of enzymes of the electron transport chain, and higher oxygen consumption rates (*Lemma et al., 2016*). The evidence collectively point towards OXPHOS as a main bioenergetic source for osteoclast differentiation, and can therefore be considered an integral aspect of the process. As a consequence of increased mitochondrial biogenesis, ROS would similarly be elevated due to the 'leakiness' of the electron transport chain, which in turn contributes to the ROS that promotes OC differentiation. As such, an increase in ROS has been associated with mitochondria biogenesis in osteoclasts (*Ishii et al., 2009*), and mitochondrial ROS is essential for osteoclast formation (*Bartell et al., 2014*). Therefore, the analysis of OXPHOS and glycolytic patterns should not be discounted in a study that focuses on redox biology. To address this concern, we evaluated the molecular subsets in the proteome involved in the energy metabolism pathways including glycolysis, tricarboxylic acid (TCA) cycle, and OXPHOS pathways (*Figure 3—figure supplement 1* and *Figure 3—source data 2*).

The loss of Rgs12 in BMMs led to the inhibition of RANKL-dependent ROS, which we demonstrated was a function of increased Nrf2 activity and elevated levels of its target antioxidant enzymes. At the same time, the proteomics analysis identified a reduction of proteins involved in the TCA cycle and OXPHOS (*Figure 3—source data 2*). Interestingly, cytochrome b5 type A (CYB5A) was against the general trend and was significantly increased by ~30% in Rgs12-deficient BMMs. The TCA cycle is responsible for the consumption of acetyl-CoA to generate NADH which is fed into the OXPHOS pathway to power ATP synthase and catalyze the formation of ATP. The impediment of OXPHOS, which requires molecular oxygen as co-factor, can shift ATP production towards anaerobic glycolysis under hypoxic and other conditions. However, the proteomic analysis revealed that protein levels of phosphofructokinase (PFK), the rate-limiting enzyme of glycolysis, were unaffected by Rgs12 deletion (*Figure 3—figure supplement 1* and *Figure 3—source data 2*). In fact, all enzymes we detected that are involved in the glycolytic pathway with the exception of enolase 3 (a.k.a. phosphopyruvate hydratase) were unaffected by the loss of Rgs12.

On one hand, the loss of Rgs12 was associated with reduced OXPHOS, which would theoretically contribute to lower ROS levels. Conversely, the inhibition of OC differentiation could lead to reduced mitochondrial biogenesis, which could also explain lowered ROS levels in Rgs12-deficient BMMs. Therefore, mitochondrial ROS could be a contributing factor in Rgs12 function—albeit likely as a secondary effect. Although OXPHOS enzymes were reduced in Rgs12-deficient osteoclasts, only 19/83 proteins identified (22.9%) were altered, as compared to 26/26 proteins (100%) identified as Nrf2 targets (*Figure 3—figure supplement 1* and *Figure 3—source data 2*). Additionally, the magnitude of protein expression fold-changes was markedly higher in Nrf2 response genes as compared to OXPHOS genes. Therefore, the higher degree of intra-group concordance and magnitude of expression of Nrf2 genes both indicate a more targeted response, whereas the weak pattern of expression of OXPHOS genes suggests an equivocal effect. Secondly, Nrf2 antioxidant proteins act downstream of mitochondrial ROS formation by acting directly on ROS. This effect is best exemplified by the fact that the ectopic expression of mitochondria-targeted catalase was able to suppress ROS and inhibit OC differentiation (*Bartell et al., 2014*). In the case of Rgs12 overexpression, it could be expected that increased mitochondrial ROS production would also need the concomitant suppression of Nrf2 to raise the overall intracellular ROS level. In the current work, we demonstrate that Rgs12 could promote Nrf2 degradation. Therefore, the inhibition of RANKL-dependent ROS associated with loss of Rgs12 was unlikely to be the direct result of glycolytic shift or reduced OXPHOS. While changes in mitochondrial ROS could be an minor aspect of Rgs12 function, but Nrf2 remains an essential component in the overall scheme.

In conclusion, we identified a new gene that could modulate the Nrf2-proteasome axis, which forms the crux of redox homeostasis in many biological contexts. Our study also points to a novel role of Rgs12 in OC redox biology, thus forming the molecular basis for developing therapies to modulate ROS for osteoporosis and other diseases of bone loss. Outside of its role in bone homeostasis, Rgs12 has also been shown to hold diverse and significant roles in other clinical contexts including pathological cardiac hypertrophy (*Huang et al., 2016*), tumor suppression in African American prostate cancer (*Wang et al., 2017*), and psychostimulant-induced increases in dopamine levels in the brain (*Gross et al., 2018*). We speculate that the ability of Rgs12 to regulate Nrf2 could be an important clue to better understand the role of Rgs12 in the context of cardiac disease and

tumorigenesis, in which ROS is known to have significant implications. Therefore, our research would have widespread appeal to audiences of different research backgrounds.

# Materials and methods

## Key resources table

| Reagent type (species) or resource | Designation | Source or reference | Identifiers | Additional information |
|---|---|---|---|---|
| Genetic reagent (*M. musculus*) | *Rgs12*<sup>flox/flox</sup> Rgs12$^{flox/flox}$ | *Yang et al., 2013* | | |
| Genetic reagent (*M. musculus*) | *LyzM*$^{Cre}$ | Jackson Laboratory | Stock #: 018956 | |
| Genetic reagent (*M. musculus*) | *Rgs12* cDNA | This paper | NCBI: NM_173402.2 | |
| Cell line (*M. musculus*) | RAW264.7 | American Type Culture Collection | Cat. #: TIB-71 | |
| Cell line (*M. musculus*) | CMG14-12 | PMID: 10934646 | | Dr. Sunao Takeshita (Nagoya City University, Nagoya, Japan) Cell line used to produce M-CSF-containing supernatant. |
| Cell line (*E. coli*) | Modified Origami B(DE3) | *Li et al., 2016a* | | Dr. Ding Xu (University at Buffalo, Buffalo, NY, USA). Modified bacterial cell line co-expresses chaperone proteins. |
| Transfected construct (synthesized) | p3XFLAG-myc-CMV-26 | Sigma-Aldrich | Cat. #: E7283 | |
| Transfected construct (synthesized) | p3XFLAG-myc-CMV-26-Rgs12 | This paper | | See Methods for details. |
| Transfected construct (synthesized) | pcDNA3.1 (+)-c-His | Genscript | | Custom vector available through Genscript's cloning services. |
| Transfected construct (synthesized) | pcDNA3.1(+)-Rgs12-c-His | This paper | | See Methods for details. |
| Recombinant DNA reagent (synthesized) | mRANKL-His (K158-D316) | Other | | Dr. Ding Xu (University at Buffalo, Buffalo, NY, USA). |
| Sequence-based reagent | Primers | Integrated DNA Technologies | | Primer sequences detailed in Methods. |
| Peptide, recombinant protein | M-CSF | R and D Systems | Cat. #: 416 ML-010 | |
| Commercial assay or kit | Acid Phosphatase, Leukocyte (TRAP) Kit | Sigma-Aldrich | Cat. #: 387A-1KT | |
| Commercial assay or kit | SimpleSeq DNA Sequencing | Eurofins Genomics | | |
| Commercial assay or kit | Pierce High Capacity Endotoxin Removal Resin | Thermo Fisher Scientific | Cat. #: 88270 | |
| Commercial assay or kit | Osteo Assay Surface | Corning | Cat. #: 3987 | |

*Continued on next page*

*Continued*

| Reagent type (species) or resource | Designation | Source or reference | Identifiers | Additional information |
|---|---|---|---|---|
| Commercial assay or kit | TRIzol Reagent | Invitrogen | Cat. #: 15596026 | |
| Commercial assay or kit | RNA to cDNA EcoDry Premix | Clontech | Cat. #: 639549 | |
| Commercial assay or kit | 2x SYBR Green qPCR Master Mix | Bimake | Cat. #: B21203 | |
| Commercial assay or kit | Rac1 Pulldown Activation Assay Kit | Cytoskeleton | Cat. #: BK035-S | |
| Chemical compound, drug | Calcein | Sigma-Aldrich | Cat. #: C0875 | |
| Chemical compound, drug | FuGENE HD Transfection Reagent | Promega | Cat. #: E2311 | |
| Chemical compound, drug | Geneticin (G418) | Thermo Fisher Scientific | Cat. #: 10131035 | |
| Chemical compound, drug | DCFDA | Sigma-Aldrich | Cat. #: D6883 | |
| Chemical compound, drug | Phenol red-free MEM | Gibco/Thermo Fisher | Cat. #: 51200038 | |
| Chemical compound, drug | cOmplete, Mini, EDTA-free | Roche/Thermo Fisher | Cat. #: 5892791001 | |
| Chemical compound, drug | Image-iT FX signal enhancer | Thermo Fisher Scientific | Cat. #: I36933 | |
| Chemical compound, drug | DAPI | Thermo Fisher Scientific | Cat. #: D1306 | |
| Chemical compound, drug | ProLong Gold Antifade Mountant | Thermo Fisher Scientific | Cat. #: P36930 | |
| Chemical compound, drug | tBHP | Sigma-Aldrich | Cat. #: B2633 | |
| Chemical compound, drug | MG-132 | Selleck Chemicals | Cat. #: S2619 | |
| Chemical compound, drug | NAC | Sigma-Aldrich | Cat. #: A9165 | |
| Chemical compound, drug | tBHQ | Sigma-Aldrich | Cat. #: 112941 | |
| Antibody | Nrf2 (H-300), rabbit polyclonal | Santa Cruz Biotechnology | Cat. #: sc-13032 | ICC (1:10), WB (1:100) |
| Antibody | Nrf2 (C-20), rabbit polyclonal | Santa Cruz Biotechnology | Cat. #: sc-722 | WB (1:100) |
| Antibody | Keap1 (E-20), goat polyclonal | Santa Cruz Biotechnology | Cat. #: sc-15246 | WB (1:100) |

*Continued on next page*

*Continued*

| Reagent type (species) or resource | Designation | Source or reference | Identifiers | Additional information |
|---|---|---|---|---|
| Antibody | Phospho-p38 (Thr180/Tyr182), rabbit polyclonal | Cell Signaling Technology | Cat. #: 9211 | WB (1:1000) |
| Antibody | p38, rabbit polyclonal | Cell Signaling Technology | Cat. #: 9212 | WB (1:1000) |
| Antibody | Phospho-p44/42 MAPK (Erk1/2) (Thr202/Tyr204) (D13.14.4E) XP Rabbit mAb | Cell Signaling Technology | Cat. #: 4370S | WB (1:1000) |
| Antibody | ERK1/2, rabbit polyclonal | Cell Signaling Technology | Cat. #: 9102 | WB (1:1000) |
| Antibody | Phospho-NFκB p65 (Ser536), rabbit monoclonal | Cell Signaling Technology | Cat. #: 3033 | WB (1:1000) |
| Antibody | NFκB p65, rabbit polyclonal | Cell Signaling Technology | Cat. #: 3034 | WB (1:1000) |
| Antibody | β-actin, mouse monoclonal | Santa Cruz Biotechnology | Cat. #: sc-47778 | WB (1:4000) |
| Software, algorithm | OsteoMeasure | OsteoMetrics | | |
| Software, algorithm | ImageJ | NIH | RRID: SCR_003070 | |
| Software, algorithm | Primer-BLAST | NIH | | |
| Software, algorithm | CFX Maestro | Bio-Rad | | |
| Software, algorithm | IonStar | *Shen et al., 2018*; *Shen et al., 2017* | | |
| Software, algorithm | Ingenuity Pathway Analysis | Qiagen | | |

## Generation of Rgs12 conditional knockout mice

*Rgs12*<sup>flox/flox</sup> mice were crossed with *LyzM*<sup>Cre</sup> transgenic mice to generate Rgs12 cKO mice specific to the myeloid lineage in a C57BL/6J background. The methodology for generating *Rgs12*<sup>flox/flox</sup> and *LyzM*<sup>Cre</sup> mice and genotyping are previously described (*Clausen et al., 1999*; *Yang et al., 2013*; *Yuan et al., 2015*). Mice used for experiments were 10–12 weeks old, as indicated. All animal studies were approved by the University at Buffalo and University of Pennsylvania Institutional Animal Care and Use Committees (IACUC).

## Quantitative Micro-CT measurements

Quantitative analysis of bone morphology and microarchitecture was performed using a micro-CT system (USDA Grand Forks Human Nutrition Research Center, Grand Forks, ND, USA). Fixed femur from 10-week-old Rgs12 cKO and control mice were analyzed and 3D reconstruction was used to determine bone volume to tissue volume (BV/TV), structure model index (SMI), trabecular thickness (Tb.Th, μm), trabecular number (Tb.N,/mm), and trabecular separation (Tb.Sp, μm).

## Bone histology analysis

Mouse tibiae and femurs from 10-week-old mice were excised and fixed in 4% PFA for 24 hr and decalcified in a 10% EDTA for 3–4 weeks at 4℃. The samples were embedded in paraffin, sectioned at 8 μm, stained with hematoxylin and eosin (H and E) or tartrate-resistant acid phosphatase (TRAP)

using the Acid Phosphatase, Leukocyte (TRAP) Kit (Sigma-Aldrich, St. Louis, MO, USA), and imaged using a Leica inverted microscope (DMI6000B, Leica, Germany).

## Dynamic histomorphometry

To assess the rate of bone formation, mice were intraperitoneally injected with calcein (25 mg/kg) twice at postnatal day 90 and day 96. Mice were euthanized and harvested 2 days after last injection. Femurs were dissected and fixed in 4% PFA for 24 hr, dehydrated in ethanol, and embedded in optimal cutting temperature (OCT) compound for cryosection without decalcification. By applying Cryofilm tape (Section Lab, Hiroshima, Japan), 8 µm longitudinal sections were cut from distal femurs using a microtome (CM1950, Leica, Germany). Measurements of surface-based histomorphometric indices were performed using the OsteoMeasure analysis system (OsteoMetrics, Decatur, GA). These indices were used to calculate bone formation rate per bone surface (BFR/BS, $\mu m^3/um^{-2}$ per day), mineral apposition rate (MAR, µm per day), OB number per bone perimeter (N.Ob/B.Pm, $mm^{-1}$) and OC number per bone perimeter (N.Oc/B.Pm, $mm^{-1}$) as previously described (*Yuan et al., 2016*; *Li et al., 2019*).

## Generation of Rgs12 expression vectors

Full length *Rgs12* (Accession: NM_173402.2) cDNA was cloned into the p3XFLAG-myc-CMV-26 expression vector (Sigma-Aldrich, St. Louis, MO, USA). Briefly, *Hind*III sites were incorporated into both termini of the *Rgs12* cDNA using restriction-site-generating PCR, and the restriction sites were used to insert the Rgs12 sequence into the expression vector containing an N-terminus FLAG tag sequence (Flag-Rgs12). The primer walking method was used to validate the correct directionality of the insert. Additionally, a vector expressing C-terminus His-tagged Rgs12 (Rgs12-His) was generated by subcloning the *Rgs12* cDNA into the pcDNA3.1(+)-c-His vector (Genscript, Piscataway, NJ, USA). *Hind*III and *Eco*RV sites were introduced by PCR and the restriction sites were used to insert *Rgs12* into the pcDNA3.1(+)-c-His vector. All vector constructs were confirmed by DNA sequencing (Eurofins Genomics, Louisville, KY, USA).

## Stable transfection

RAW264.7 cells were purchased from ATCC which was confirmed to be free of mycoplasma contamination. RAW264.7 cells were seeded at $2 \times 10^6$ cells per 6-well and transfected using FuGENE HD reagent (Promega, Madison, WI, USA) according to manufacturer's instructions at a 1:3 DNA to transfection reagent ratio. After 48 hr post-transfection, cells were treated with 0.4 mg/mL geneticin (G418, Thermo Fisher Scientific, Waltham, MA, USA) for 2 weeks until antibiotic-resistant colonies are formed. Stably transfected cells were thereafter maintained in media containing 0.4 mg/mL G418.

## In vitro osteoclastogenesis and bone resorption assays

The vector encoding the recombinant mRANKL-His (K158-D316) construct and a modified *E. coli* strain Origami B(DE3) cells (EMD Millipore, Billercica, MA, USA) co-expressing chaperone proteins that was used to express the recombinant RANKL were generous gifts from Dr. Ding Xu at the University at Buffalo. The protocol for expressing and purifying mRANKL-His was described previously (*Li et al., 2016a*). Endotoxins were removed using the Pierce High Capacity Endotoxin Removal Resin (Thermo Fisher Scientific, Waltham, MA, USA).

BMMs were obtained from the tibiae and femurs of 10-week-old mice as described previously (*Yang and Li, 2007*). For in vitro osteoclastogenesis experiments, BMMs were seeded at $2 \times 10^6$ cells per 24-well plates and stimulated with 100 ng/mL RANKL and 20 ng/mL M-CSF (R and D Systems, Minneapolis, MN, USA) for 5 days to generate mature OCs. RAW264.7 cells were seeded at $1.35 \times 10^4$ cells per 24-well and stimulated with M-CSF and RANKL for 5 days. Prior to fixing and staining, RAW264.7-derived OCs were gently but thoroughly rinsed with PBS to remove mononuclear cells that tend to obscure OCs during imaging. TRAP staining was performed using the Acid Phosphatase, Leukocyte (TRAP) kit (Sigma-Aldrich, St. Louis, MO, USA). Cells were imaged using the Cytation 5 Cell Imaging Multi-Mode Reader (BioTek, Winooski, VT, USA) using the montage function. Osteoclasts were quantified by counting the number of TRAP$^+$, multinucleated cells (MNCs,$\geq$3 nuclei/cell) per well. Average osteoclast area was determined by measuring total TRAP$^+$ area using

the ImageJ software (US National Institute of Health, Bethesda, MA, USA) and dividing the value by total osteoclast number.

For bone resorptive activity experiments, the method above was applied except that cells were seeded on Osteo Assay Surface plates (Corning, Corning, NY) and allowed to differentiate and resorb the calcium phosphate surface for 5–6 days. A 10% bleach solution was added to each well for 5 min to remove cells, followed by rinsing with deionized water, and air-dried. Resorption pits were visualized and captured under a light microscope (DMI6000B, Leica, Germany) and analyzed using ImageJ software as previously described (*Li et al., 2016b*).

## Reverse transcription and quantitative PCR

Total RNA was isolated from cultured BMMs and OCs using Trizol reagent (Invitrogen, Carlsbad, CA, USA) following manufacturer's instructions. cDNA was reverse transcribed from 2 µg total RNA using the RNA to cDNA EcoDry Premix kit (Clontech, Palo Alto, CA, USA). Primers were designed using Primer-BLAST (*Ye et al., 2012*) and obtained from IDT (Integrated DNA Technologies, San Diego, CA, USA). Rgs12 (F: 5'-AAGATCCATTCCCTAGTGACC-3', R: 5'-ACCTCCACTTTCCCACCC TG-3', 587 bp), Nrf2 (F: 5'-GCCCACATTCCCAAACAAGAT-3', R: 5'-CCAGAGAGCTATTGAGGGAC TG-3', 172 bp), Keap1 (F: 5'-TGCCCCTGTGGTCAAAGTG-3', R: 5'-GGTTCGGTTACCGTCCTGC-3', 104 bp), β-actin (F: 5'-CTAGGCACCAGGGTGTGAT-3', R: 5'-TGCCAGATCTTCTCCATG TC-3', 148 bp), GAPDH (F: 5'-AGGTCGGTGTGAACGGATTTG-3', R: 5'-TGTAGACCATGTAGTTGAGGTCA-3', 123 bp). qPCR was performed using the 2x SYBR Green qPCR Master Mix following manufacturer's instructions (Bimake, Houston, TX, USA). All reactions were performed in triplicate and normalized to the housekeeping gene β-actin or GAPDH as indicated. Data analysis was performed using the CFX Maestro software (Bio-Rad, Hercules, CA, USA).

## Rac1-GTP immunoprecipitation

The Rac1-GTP pulldown assay was performed following manufacturer instructions in the Rac1 activation assay kit (Cytoskeleton, Denver, CO, USA).

## ROS measurement

To measure ROS production, BMMs were seeded into black, glass-bottom 96-well plates and cultured with M-CSF for 48 hr until confluence. Cells were loaded with 20 µM 2'7'-dichlorofluorescein diacetate (DCFDA, Sigma) at 37°C for 30 min and washed using PBS. The cells were swapped into complete phenol red-free MEM (Gibco) containing RANKL/M-CSF. Fluorescence intensity was measured using the Cytation five plate reader (BioTek) with excitation wavelength at 488 nm and emission wavelength at 535 nm. Background signals (cells not loaded with DCF-DA) were subtracted. Experiments were carried out in quintuplicate wells.

## Protein extraction and precipitation/On Pellet Digestion

Cells were harvested using ice-cold lysis buffer (50 mM Tris-formic acid, 150 mM NaCl, 0.5% sodium deoxycholate, 1% SDS, 2% NP-40, pH 8.0) with protease inhibitor (cOmplete, Mini, EDTA-free; Roche, Mannheim, Germany). Samples were prepared for MS analysis using an established method (*Shen et al., 2018*; *Shen et al., 2017*).

## Liquid Chromatography-Tandem mass spectrometry analysis

The 'IonStar' LC-MS experimental pipeline was developed and optimized in a previous study (*Shen et al., 2018*; *Shen et al., 2017*). A stringent set of criteria including a low peptide and protein false discovery rate (FDR) of <1% and ≥2 peptides per protein was used for protein identification. An ion current-based quantification method (IonStar processing pipeline) was described previously (*Shen et al., 2018*; *Shen et al., 2017*).

## Bioinformatics analysis

Ingenuity Pathway Analysis (Qiagen, Redwood City, CA, USA) was used to perform gene ontology enrichment analysis. Hierarchical clustering analysis and heat map visualizations were performed using the *agnes* function in R Package *cluster* and *ggplot2* with the *viridis* color palette, respectively.

## Immunofluorescence

For the Nrf2 nuclear translocation experiment, BMMs were cultured on coverslips and treated with RANKL and M-CSF for 72 hr, 5 mM NAC for 16 hr, or 50 μM tBHP for 16 hr. Coverslips were fixed with 4% paraformaldehyde solution in PBS for 10 min at room temperature and permeabilized using 0.1% Triton X-100 for 5 min at room temperature. Coverslips were blocked using Image-iT FX signal enhancer (Thermo Fisher Scientific) for 1 hr at room temperature, stained with the primary antibody in 1% BSA/TBST overnight at 4°C, and stained with the secondary antibody for 1 hr at room temperature. 4,6-diamidino-2-phenylindole (DAPI) (Sigma) was used as a counterstain for nuclei. The coverslips were mounted using ProLong Gold antifade mountant (Thermo) and images were obtained using a fluorescence microscope (Leica, Wetzlar, Germany).

## Western blotting

For experiments studying the Keap1-Nrf2 pathway, cells were cultured in 6-well plates and pretreated with the indicated concentrations of tBHQ or 25 μM MG-132 for 4 hr. For MAPK and NFκB activation experiments, stable-transfected RAW264.7 cells were cultured in 6-well plates and starved in serum-free medium containing 5 mM NAC for 16 hr. Cells were subsequently induced with RANKL (200 ng/mL) and M-CSF (100 ng/mL) for the indicated times. Western blotting was performed as described previously (*Yuan et al., 2016*). The primary antibodies used in this study were as follows: Nrf2 (H-300) and Nrf2 (C-20) (1:100, Santa Cruz Biotechnology, Dallas, TX, USA), Keap1 (E-20) (1:100, SCBT), phospho-p38 (Thr180/Tyr182) (1:1000, Cell Signaling Technology), p38 (1:1000, CST), phospho-p44/42 MAPK (Erk1/2) (Thr202/Tyr204) (1:1000, CST), ERK1/2 (1:1000, CST), phospho-NFκB p65 (Ser536) (1:1000, CST), NFκB p65 (1:1000, CST), and β-actin (1:4000, SCBT). Densitomety analysis was performed using ImageJ (*Schindelin et al., 2012*) and normalized to the β-actin signal. Relative phosphorylation of was presented as the ratio between the phosphorylated normalized to the non-phosphorylated/total protein. NAC, tBHQ, and tBHP were obtained from Sigma-Aldrich (St. Louis, MO, USA), and MG-132 was obtained from Selleck Chemicals (Houston, TX, USA).

## Acknowledgements

This work was supported by the National Institute of Arthritis and Musculoskeletal and Skin Diseases (NIAMS, AR066101), the National Institute of Aging (NIA, AG048388) awarded to Dr. Shuying Yang, and by grants from the Penn Center for Musculoskeletal Disorders (NIH/NIAMS, P30-AR069619). We are deeply appreciative for the UPenn histology core for their assistance in cryosectioning processes and Osteomeasure analysis. We would also like to thank Dr. Ding Xu and Dr. Sunao Takeshita for their generous gifts of the mRANKL-His (K158-D316) vector and CMG14-12 cell line, respectively.

# Additional information

### Funding

| Funder | Grant reference number | Author |
| --- | --- | --- |
| National Institute of Arthritis and Musculoskeletal and Skin Diseases | R01AR066101 | Shuying Yang |
| National Institute on Aging | R01AG048388 | Shuying Yang |

The funders had no role in study design, data collection and interpretation, or the decision to submit the work for publication.

### Author contributions

Andrew Ying Hui Ng, Conceptualization, Data curation, Software, Formal analysis, Investigation, Visualization, Writing—original draft, Writing—review and editing; Ziqing Li, Megan M Jones, Data curation, Investigation; Shuting Yang, Data curation, Investigation, Methodology; Chunyi Li, Data curation, Formal analysis, Validation, Investigation, Methodology; Chuanyun Fu, Data curation, Methodology; Chengjian Tu, Software, Supervision, Validation, Methodology; Merry Jo Oursler, Resources, Software, Investigation, Writing—review and editing; Jun Qu, Resources, Software,

Supervision, Funding acquisition, Methodology; Shuying Yang, Conceptualization, Resources, Supervision, Funding acquisition, Investigation, Project administration, Writing—review and editing

### Author ORCIDs
Andrew Ying Hui Ng  https://orcid.org/0000-0001-7771-6574
Ziqing Li  https://orcid.org/0000-0003-3596-3520
Shuying Yang  https://orcid.org/0000-0002-7126-6901

### Decision letter and Author response
Decision letter https://doi.org/10.7554/eLife.42951.019
Author response https://doi.org/10.7554/eLife.42951.020

## Additional files

### Supplementary files
• Transparent reporting form
DOI: https://doi.org/10.7554/eLife.42951.017

### Data availability
All data generated or analysed during this study are included in the manuscript and supporting files. Source data files have been provided for Figures 1 and 3.

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
