## [Decision Letter]

Thank you for sending your article entitled "Rgs12 enhances osteoclastogenesis by suppressing Nrf2 activity and promoting the formation of reactive oxygen species" for peer review at *eLife*. Your article is being evaluated by three peer reviewers, one of whom is a member of our Board of Reviewing Editors, and the evaluation is being overseen by Marianne Bronner as the Senior Editor.

Given the list of essential revisions, including new experiments, the editors and reviewers invite you to respond within the next two weeks with an action plan and timetable for the completion of the additional work. We plan to share your responses with the reviewers and then issue a binding recommendation.

Summary:

All three reviewers agree that demonstration of Rgs12 deletion led to increased bone mass and its overexpression increased OC number and size were novel observations. The proteomic analysis of Rgs12-deficient OCs identified novel peptides and upregulation of antioxidant enzymes under the transcriptional regulation of Nrf2 and is another strong component of the manuscript. Conversely, by showing that Rgs12 overexpression suppressed Nrf2 through a mechanism dependent on the 26S proteasome and promoted RANKL-induced phosphorylation of ERK1/2 and NFκB, a putative mechanism for the in vivo effects has been provided.

Nevertheless, there were four major concerns that need to be addressed.

Essential revisions:

First, the authors need to demonstrate in vitro functional activities of the osteoclast including resorption on dentin; this would further strengthen their argument of a major osteoclast defect as the cause of high bone mass with Rgs12 deletion.

Second, dynamic histomorphometry from tibia/femur and or vertebrae is essential to support the in vivo mechanisms proposed in the paper.

Third, further in vitro studies are needed in order to determine the differential response of bone marrow cells to mCSF and RANKL for levels of Rgs12 mRNA and protein compared to cultures treated with M-CSF alone.

Finally, no experiments were performed to test whether the altered Nrf2 levels cause the phenotype induced by Rgs12 deletion: these need to be done.

In addition, some consideration should be included about the effects of excess ROS on the metabolic program of the osteoclast.

*Reviewer #1:*

This manuscript reports that deletion of Rgs12 in vivo increased bone mass and in vitro decreased osteoclastogenesis and ROS production and increased Nrf2 protein level and Nrf2-dependent antioxidant enzymes.

1) The major conclusion of the paper, that Rgs12 deletion decreases osteoclast formation, needs to be tested in vivo by reporting quantitation of osteoclast and osteoblast numbers and activity in the histology shown in Figure 1C. The in vivo data also lacks an important control (LysM-cre;Rgs12+/+ mice) because cre expression can cause effects on bone phenotype even without the presence of a floxxed allele. The in vitro data also needs to be strengthened by measuring osteoclast activity (pit formation) rather than relying on osteoclast number and size.

2) The effects of Rgs12 deletion and overexpression on Rgs12 protein levels appear to be quite modest and not observable in the western blots, and their quantitation by densitometry is unlikely to be sensitive enough to reliably detect the small differences, which, even if true, are unlikely to be sufficient to have a major effect on the cellular phenotype. Moreover, no experiments were performed to test whether the altered Nrf2 levels cause the phenotype induced by Rgs12 deletion.

*Reviewer #2:*

In this manuscript, using the LysM-Cre transgenic line or overexpressing the gene in RAW264.7 cells, Ng et al. showed that Rgs12 deletion led to increased bone mass whereas its overexpression increased OC number and size. Proteomic analysis of Rgs12-deficient OCs identified an upregulation of antioxidant enzymes under the transcriptional regulation of Nrf2, the master regulator of oxidative stress. They confirmed an increase of Nrf2 activity and impaired production of ROS in Rgs12- deficient cells. Conversely, Rgs12 overexpression suppressed Nrf2 through a mechanism dependent on the 26S proteasome, and promoted RANKL-induced phosphorylation of ERK1/2 and NFκB, which was abrogated by antioxidant treatment. In sum, the authors demonstrated a potentially novel upstream network for regulating osteoclastogenesis via the ROS pathway.

The authors have provided more insights into the role ROS plays in regulating osteoclastogenesis, clearly an important pathway in the differentiation process. The paper is well written and there is some novelty particularly in the mechanism through which Rgs12 mediates suppression of Nrf2 and that in turn, down regulates ROS via anti-oxidant factors. There are however some concerns (labelled 1-6) which are major and must be addressed; even though enthusiasm is moderate, more work is required to firmly establish the role of Rgs12 in osteoclast differentiation.

1) The mechanism underlying attenuated osteoclastogenesis in the Rgs12 deletion model is presumed to be an upregulation of the Nrf2 anti-oxidant system. Yet it is unclear why Keap1 levels are high in the Rgs12 null cells particularly since Keap1 sequesters Nrf2 and prevents its action. A more comprehensive approach to understanding that feature beyond a theoretical compensatory response is needed to fully delineate the pathway's importance for osteoclastogenesis.

2) On a similar note, it is unclear why Nrf2 protein levels are not increased in the proteomic analysis of the Rgs12 deleted cells, particularly since the downstream anti-oxidant proteins are increased, one would have expected those changes. Did the authors measure Nrf2 protein by WB in the Rgs12 null osteoclasts?

3) The conditional deletion by LysMCre of Rgs12 led to high bone mass and presumed osteoclast dysfunction. The BMM cells during differentiation showed less nuclei and lower number of OCs; however the authors did not perform studies on the functional significance of this osteoclast phenotype; did the osteoclasts from the LysMCre mice resorb less bone in vitro i.e. on a bone matrix?

4) In that same vein, there was no dynamic histomorphometry studies to assess the etiology of the high bone mass. An obvious conclusion would be that there is less resorption. However it would be important to have measured eroded surface, bone formation rates and other parameters that could be responsible for the high bone mass in the LysMCre Rgs12 deficient mice. Nor it should be noted were osteoclast numbers counted by static measurements. This needs clarification particularly if osteoclast number remained the same in the Rgs12 null mice, but the cells were not functional (less nuclei) and clastokines were still being released to stimulate osteoblasts as in the catK ko mice. Clearly more insight into the in vivo phenotype would have been preferable. Similarly, resorption markers and/or an OVX experiment would have also been useful.

5) The proteomic analysis of the Rgs12 deficient osteoclasts revealed a shift toward more glycolytic pathways. This would be consistent with less ROS generation; but is it that shift that directly impairs osteoclast function? In other words, when osteoclasts become glycolytic, the cells cannot fuse or resorb bone? It is disappointing that the authors did not examine OxPhos and glycolytic patterns during OC differentiation in these cells to complete the characterization of their metabolic programs. Since the premise of the paper is the critical nature of ROS levels, defining their oxidative capacity would seem to be necessary.

6) The NAC suppression of NFκB activation is not surprising; but is it specific for ROS mediated activation or does it have other downstream activity?

*Reviewer #3:*

In this manuscript the authors examine the role that Rgs12 has in osteoclast formation and function. They find that targeted deletion of Rgs12 in myeloid cells produces a phenotype in mice of increased bone mass. In vitro studies of BMMs demonstrates that Rgs12 deletion decreased the number of osteoclasts that formed in vitro with 6 or greater nuclei with no effect on the number of osteoclasts that had 3 to 5 nuclei. Conversely, overexpression of Rgs12 in RAW264.7 cells increased the overall size of osteoclasts that formed in vitro and the number of osteoclasts with greater than 10 nuclei. Overexpression of Rgs12 also decreased the number of osteoclasts that had less than 10 nuclei in the RAW cell cultures.

Proteomic analysis identified Nrf2-mediated stress responses as potentially mediating the effects of Rgs12 deletion on osteoclastogenesis in vitro. Consequently, the authors studied the role that Rgs12 had in mediating responses of Nrf2 in osteoclasts. They found that deletion of Rgs12 increased levels of the transcription factor Nrf2, which is induced by oxidative stress, and its regulatory protein Keap1 in BMMs that were induced to form osteoclasts with RANKL. RANKL also increased the nuclear localization of Nrf2 in Rgs12-deleted cells, which is consistent with the decrease in osteoclastogenesis seen in Rgs12-deleted BMM cultures, since Nrf2 activation is known to inhibit osteoclast formation. RANKL did not stimulate oxidative stress in Rgs12-deleted BMM cultures, implying that the increased levels of Nrf2 stimulated an antioxidant pathway to reduce ROS levels.

Overexpression of Rgs12 in RAW cells reduced the ability of a Keap2 inhibitor (tBHQ) to induce Nrf2. To better understand how this might be occurring the authors examined the effects that the proteasome inhibitor, MG-132, had on Nrf2 levels in RAW cells in the presence or absence of Rgs12 overexpression. They found that in the absence of proteasome degradation, overexpression of Rgs12 did not inhibit Nrf2 protein levels but rather enhanced them suggesting that Rgs12 regulates interactions of the proteasome degradation pathway with Nrf2.

Finally, the authors demonstrated that pretreating RAW cells, which overexpressed Rgs12, with an antioxidant (to decrease ROS) inhibited ERK and NF-κB activation by RANKL. They conclude that this data implies that Rgs12 promotes ROS production as one mechanism for its ability to stimulate osteoclast formation and function.

This is a complex subject and the authors have attempted a variety of investigations to support their conclusions. In general, they have been successful but there are several criticisms that they need to address.

1) The in vivo model of targeted deletion of Rgs12 in myeloid cells is lacking any evidence that this manipulation resulted in a decrease in osteoclast number or function in the mice. The authors need to perform histomorphometric studies to confirm that the increase in bone mass that they find with targeted Rgs12 deletion is associated with a decrease in osteoclast number, size or erosion surface. Studies of osteoblasts and bone formation rate are also needed.

2) What is the effect of 1 to 5 days of RANKL treatment on levels of Rgs12 mRNA and protein in BMM cultures compared to cultures treated with M-CSF alone? This is a fundamental experiment that the authors do not present. Are levels increased, unchanged or decreased by RANKL? The authors seem to be arguing that RANKL increases Rgs12 levels since overexpression of this protein enhanced and deletion decreased RANKL-induced osteoclastogenesis. However, the authors really need to show this kind of data to confirm this hypothesis.

[Editors' note: further revisions were requested prior to acceptance, as described below.]

Thank you for resubmitting your article "Regulator of G protein signaling 12 enhances osteoclastogenesis by suppressing Nrf2-dependent ROS generation" for consideration by *eLife*. Your article has been reviewed by two peer reviewers, and the evaluation has been overseen by a Reviewing Editor and Marianne Bronner as the Senior Editor. The reviewers have opted to remain anonymous.

The reviewers have discussed the reviews with one another and the Reviewing Editor has drafted this decision to help you prepare a revised submission.

The authors have responded to most of the comments and the paper is more comprehensive. We would like to move forward however, there are two sticking points of concern that you must address more completely:

1) Figure 4A western is not at all convincing that deletion of Rgs12 increases Nrf2 activity and this is a major premise of the paper – a critical figure; we would like to see the difference more clearly.

2) The authors did NOT respond to the issue of the proper control for the LysMCre Rgs12 targeted deletion by skeletal phenotyping; the appropriate control is the LysMCreRgs12+/+, that should be relatively easy to phenotype and exclude a transgenic Cre effect. All that you say is that the animals are not different in size. It should be relatively straightforward since these are the controls to have bone density data.

---

## [Author Response]

[Editors' note: the authors’ plan for revisions was approved and the authors made a formal revised submission.]

Reviewer #1:This manuscript reports that deletion of Rgs12 in vivo increased bone mass and in vitro decreased osteoclastogenesis and ROS production and increased Nrf2 protein level and Nrf2-dependent antioxidant enzymes.1) The major conclusion of the paper, that Rgs12 deletion decreases osteoclast formation, needs to be tested in vivo by reporting quantitation of osteoclast and osteoblast numbers and activity in the histology shown in Figure 1C. The in vivo data also lacks an important control (LysM-cre;Rgs12+/+ mice) because cre expression can cause effects on bone phenotype even without the presence of a floxxed allele. The in vitro data also needs to be strengthened by measuring osteoclast activity (pit formation) rather than relying on osteoclast number and size.

A) Quantitation of osteoclast and osteoblast numbers

Findings: Quantification of bone cells on femoral bone sections obtained from Rgs12fl/fl and LysM;Rgs12fl/fl mice show a significant decrease in osteoclast numbers as TRAP+ multinucleated cells and no changes to osteoblast numbers.

Manuscript revisions:

- Experiments completed and incorporated into Figure 1 as panels J-M.

- Data used to update Results section.

- Discussion updated.

- New methodology added to Methods section.

B) Measuring the rate of bone formation by dynamic histomorphometric analysis

Findings: Dynamic histomorphometric analysis by double calcein labeling showed no difference mineral apposition rate (MAR) nor bone formation rate (BFR) between Rgs12^fl/fl^ and LysM;Rgs12^fl/fl^ mice.

Manuscript revisions:

- Experiments completed and data incorporated into Figure 1 as panels N-P.

- Data used to update Results section.

- Discussion updated.

- New methodology added to Methods section.

C) Assessing the effects of Rgs12 deletion or overexpression on osteoclastic bone resorption

Findings: Rgs12-deleted osteoclast precursors cultured ex vivo showed significantly reduced bone resorption activity whereas Rgs12 overexpressed in RAW264.7 cells increased bone resorption.

Manuscript revisions:

- Experiments completed and incorporated into Figure 2 as panels E-F and K-L.

- Data used to update Results section.

- Discussion updated.

- New methodology added to Methods section.

D) LysM control

Cre expression can lead to phenotypes independent of the targeted gene, including skeletal phenotypes.^7^ Particularly, we have noted this phenomenon in mice with osterix promoter-driven Cre, which are dramatically smaller compared to their wild-type control, which has been documented by Huang and Oslen.^8^ On the contrary, we did not observe any differences in the size of LysM;Rgs12^+/+^ mice compared to the Rgs^fl/fl^ or wild type mice. An in-depth search of publications did not yield any reports of skeletal defects associated with the LysM-Cre transgenic line. Moreover, many published work using the LysM-Cre system also use floxed mice as control, and did not include the LysM-Cre;gene^+/+^ control.^9-14^ Therefore, it unlikely that LysM-Cre has a significant impact on the skeletal phenotype.

2) The effects of Rgs12 deletion and overexpression on Rgs12 protein levels appear to be quite modest and not observable in the western blots, and their quantitation by densitometry is unlikely to be sensitive enough to reliably detect the small differences, which, even if true, are unlikely to be sufficient to have a major effect on the cellular phenotype. Moreover, no experiments were performed to test whether the altered Nrf2 levels cause the phenotype induced by Rgs12 deletion.

The deletion of Rgs12 in LysM;Rgs12^fl/fl^ BMMs resulted in approximately 70% reduction of Rgs12 transcript levels as measured by qPCR (Figure 1B). Additionally, the ectopic expression of Rgs12 in RAW264.7 cells led to an approximately doubled Rgs12 protein levels as measured by western blotting (Figure 2C).

The role of Nrf2 in osteoclasts and bone biology has been previously reported and is consistent with the findings reported in the current study. In osteoclasts, ROS is conducive towards its formation and bone resorption whereas antioxidants are inhibitory. Therefore, it should be expected that elevated Nrf2 activity and increased expression of antioxidant enzymes, as observed in Rgs12^–/–^ mice, would suppress osteoclast differentiation. Conversely, reduced Nrf2 activity should promote osteoclastogenesis. Supporting this idea, global Nrf2 knockout mice showed increased osteoclast numbers in distal femurs and increased osteoclast surface in lumbar vertebrae.^15^ Furthermore, Nrf2^–/–^ mice have reduced trabecular and cortical bone, along with a reduction in bone mechanical strength, although this may likely be attributed to reduced bone mineralization and formation by osteoblasts.^15-17^ As a confounding factor, Nrf2^–/–^ osteoblasts also have increased RANKL expression, thereby indirectly promoting osteoclast formation in Nrf2 global knockout mice.^18^ However, the direct role of Nrf2 in osteoclasts has been demonstrated. It was shown that RANKL stimulation of RAW264.7 cells lowers their Nrf2/Keap1 ratio and leads to a decrease in the expression of antioxidant response element (ARE) genes, whereas Nrf2 overexpression elevates antioxidant expression and suppresses osteoclast differentiation.^19-21^ Importantly, Nrf2 deficiency in osteoclasts promotes the RANKL-induced activation of MAPKs, c-Fos, and the subsequent induction of NFATc1.^21^ Therefore, it is clear that Nrf2 has a direct effect on osteoclasts. Whereas Nrf2 deficiencycontributes to heightened osteoclastic formation and a net bone loss phenotype,^15,18^ we observe the exact opposite in Rgs12 conditional knockout mice wherein Nrf2 activity was elevated.

Reviewer #2:[…] The paper is well written and there is some novelty particularly in the mechanism through which Rgs12 mediates suppression of Nrf2 and that in turn, down regulates ROS via anti-oxidant factors. There are however some concerns (labelled 1-6) which are major and must be addressed; even though enthusiasm is moderate, more work is required to firmly establish the role of Rgs12 in osteoclast differentiation.1) The mechanism underlying attenuated osteoclastogenesis in the Rgs12 deletion model is presumed to be an upregulation of the Nrf2 anti-oxidant system. Yet it is unclear why Keap1 levels are high in the Rgs12 null cells particularly since Keap1 sequesters Nrf2 and prevents its action. A more comprehensive approach to understanding that feature beyond a theoretical compensatory response is needed to fully delineate the pathway's importance for osteoclastogenesis.

Keap1 plays an important role for maintaining Nrf2 at low levels by acting as an adaptor protein for the Cullin-3-based ubiquitin E3 ligase, which mediates Nrf2 degradation via the 26S proteasome. While Keap1 levels appear to be elevated in Rgs12-deficient BMMs (Figure 4A, B), we also found that the Rgs12-mediated degradation of Nrf2 is independent of Keap1 function (Figure 5B, C). tBHQ (*tert*-butylhydroquinone) is a selective inhibitor of Keap1 and was shown to directly covalently modify cysteine residues on Keap1 to inhibit its activity,^22^ but it was unable to block Rgs12-mediated degradation of Nrf2. We subsequently showed that Rgs12 is likely to regulate the interactions of the proteasome degradation pathway with Nrf2. Therefore, while Keap1 certainly plays a canonical role in Nrf2 suppression, the evidence suggests that Keap1 is not involved in the Rgs12 function.

2) On a similar note, it is unclear why Nrf2 protein levels are not increased in the proteomic analysis of the Rgs12 deleted cells, particularly since the downstream anti-oxidant proteins are increased, one would have expected those changes. Did the authors measure Nrf2 protein by WB in the Rgs12 null osteoclasts?

The upregulation of Nrf2-dependent antioxidant enzymes certainly points toward increased Nrf2 protein levels. However, the transcription factor was not detected in the proteomics analysis, which is likely a result of technical factors inherent to current LC-MS capabilities. It was previously reported that transcription factor protein levels are notoriously difficult to obtain due to many having relatively low expression levels that are generally below current mass spectrometer detection limits.^23^ The quantitation of transcription factors typically entails an upstream enrichment/separation step before LC-MS analysis. Therefore, the absence of Nrf2 in our proteomics analysis is not an unusual happenstance from a technical standpoint. Alternatively, we confirmed our proteomics findings by showing increased Nrf2 protein levels and nuclear translocation activity in Rgs12-deficient osteoclasts (Figure 4A-C)

Technical reasons that could explain why Nrf2 was not detected in the LC-MS analysis as described above was added to the Discussions section.

3) The conditional deletion by LysMCre of Rgs12 led to high bone mass and presumed osteoclast dysfunction. The BMM cells during differentiation showed less nuclei and lower number of OCs; however the authors did not perform studies on the functional significance of this osteoclast phenotype; did the osteoclasts from the LysMCre mice resorb less bone in vitro i.e. on a bone matrix?

We assessed the effects of Rgs12 deletion on osteoclastic bone resorption. Osteoclasts derived from BMMs isolated from Rgs12^fl/fl^ and LysM;Rgs12^fl/fl^ mice was differentiated on calcium phosphate-coated plates. Resorptive activity was measured as a function of the percentage area of calcium phosphate removed relative to the total area.

Findings: Rgs12-deleted osteoclast precursors cultured ex vivo showed significantly reduced bone resorption activity whereas Rgs12 overexpressed in RAW264.7 cells increased bone resorption.

Manuscript revisions:

- Experiments completed and incorporated into Figure 2 as panels E-F and K-L.

- Data used to update Results section.

- Discussion updated.

- New methodology added to Methods section.

4) In that same vein, there was no dynamic histomorphometry studies to assess the etiology of the high bone mass. An obvious conclusion would be that there is less resorption. However it would be important to have measured eroded surface, bone formation rates and other parameters that could be responsible for the high bone mass in the LysMCre Rgs12 deficient mice. Nor it should be noted were osteoclast numbers counted by static measurements. This needs clarification particularly if osteoclast number remained the same in the Rgs12 null mice, but the cells were not functional (less nuclei) and clastokines were still being released to stimulate osteoblasts as in the catK ko m. Clearly more insight into the in vivo phenotype would have been preferable. Similarly, resorption markers and/or an OVX experiment would have also been useful.

We addressed this issue by measuring the rate of bone formation by dynamic histomorphometric analysis. 12 week-old Rgs12^fl/fl^ and LysM;Rgs12^fl/fl^ mice will be given two doses of calcein label 5-7 days apart, by intraperitoneal injection. Femurs were isolated and visualized by fluorescence imaging and the mineral apposition rate (MAR) and bone formation rate (BFR) were determined. Additionally, femoral sections were stained and bone cells were quantified.

A) Measuring the rate of bone formation by dynamic histomorphometric analysis

Findings: Dynamic histomorphometric analysis by double calcein labeling showed no difference mineral apposition rate (MAR) nor bone formation rate (BFR) between Rgs12^fl/fl^ and LysM;Rgs12^fl/fl^ mice.

Manuscript revisions:

- Experiments completed and data incorporated into Figure 1 as panels N-P.

- Data used to update Results section.

- Discussion updated.

- New methodology added to Methods section.

B) Quantitation of osteoclast and osteoblast numbers

Findings: Quantification of bone cells on femoral bone sections obtained from Rgs12fl/fl and LysM;Rgs12fl/fl mice show a significant decrease in osteoclast numbers as TRAP+ multinucleated cells and no changes to osteoblast numbers.

Manuscript revisions:

- Experiments completed and incorporated into Figure 1 as panels J-M.

- Data used to update Results section.

- Discussion updated.

- New methodology added to Methods section.

5) The proteomic analysis of the Rgs12 deficient osteoclasts revealed a shift toward more glycolytic pathways. This would be consistent with less ROS generation; but is it that shift that directly impairs osteoclast function? In other words, when osteoclasts become glycolytic, the cells cannot fuse or resorb bone? It is disappointing that the authors did not examine OxPhos and glycolytic patterns during OC differentiation in these cells to complete the characterization of their metabolic programs. Since the premise of the paper is the critical nature of ROS levels, defining their oxidative capacity would seem to be necessary.

We probed into the possible involvement of metabolic programs and redox homeostasis through literature and our proteomics data, detailed below. This important consideration incorporated into the Discussion section. Proteomics data detailing proteins in bioenergetic programs were extracted into an Excel sheet as added as Figure 3—figure supplement 1. Pathway analysis diagrams corresponding to glycolysis, TCA cycle, and OXPHOS pathways were added as Figure 3—figure supplement 2.

Hypothesis: The inhibition of RANKL-dependent ROS associated with loss of Rgs12 is not the direct result of glycolytic shift or reduced oxidative phosphorylation.

Text added to the Discussion section (eighth paragraph):

ROS is an inevitable byproduct of the mitochondrial electron transport chain (oxidative phosphorylation, OXPHOS) during ATP synthesis. […] Therefore, changes in mitochondrial ROS could certainly be an aspect of Rgs12 function, but Nrf2 remains an essential requirement in the overall scheme.

6) The NAC suppression of NFκB activation is not surprising; but is it specific for ROS mediated activation or does it have other downstream activity?

N-acetylcysteine (NAC) is a generally thought to provide the precursor substrate (_L_-cysteine) for the synthesis of glutathione (GSH), an important thiol-containing antioxidant protein.^24^ NAC has been shown in several studies to cause the inactivation of ERK, JNK, and p38 MAPK, and inhibit osteoclast formation, presumably through its antioxidant capability.^25,26^ Additionally, NAC supplementation in mice leads to decreased osteoclast differentiation and increased bone mass,^27^ which could also be substantiated by reduced RANKL production as seen in a study of IL-17-induced RANKL production in rheumatoid arthritis synovial fibroblasts.^28^ In our study, the key purpose of NAC was to show that the Rgs12-dependent phosphorylation of ERK1/2 and NFκB was dependent on ROS (Figure 6A-B). However, as with any study using small-molecule inhibitors, off-target effects are a reasonable concern. To the point of whether NAC has other functions outside of its antioxidant role, recent findings have shown that it could directly antagonize the activity of proteasome inhibitors (bortezomib, lactacystin, and MG132).^29,30^ This would of course have significant implications in proteasome-dependent proteins such as NFκB (via IκBα degradation) and Nrf2—*if* NAC was used together with proteasome inhibitors. Importantly, NAC itself is not an agonist of proteasome function.^30^ Within the context of our study, NAC alone could attenuate ERK1/2 and NFκB activation, likely as a result of its antioxidant function.

Reviewer #3:[…] This is a complex subject and the authors have attempted a variety of investigations to support their conclusions. In general, they have been successful but there are several criticisms that they need to address.1) The in vivo model of targeted deletion of Rgs12 in myeloid cells is lacking any evidence that this manipulation resulted in a decrease in osteoclast number or function in the mice. The authors need to perform histomorphometric studies to confirm that the increase in bone mass that they find with targeted Rgs12 deletion is associated with a decrease in osteoclast number, size or erosion surface. Studies of osteoblasts and bone formation rate are also needed.

A) Quantitation of osteoclast and osteoblast numbers

Findings: Quantification of bone cells on femoral bone sections obtained from Rgs12fl/fl and LysM;Rgs12fl/fl mice show a significant decrease in osteoclast numbers as TRAP+ multinucleated cells and no changes to osteoblast numbers.

Manuscript revisions:

- Experiments completed and incorporated into Figure 1 as panels J-M.

- Data used to update Results section.

- Discussion updated.

- New methodology added to Methods section.

B) Measuring the rate of bone formation by dynamic histomorphometric analysis

Findings: Dynamic histomorphometric analysis by double calcein labeling showed no difference mineral apposition rate (MAR) nor bone formation rate (BFR) between Rgs12^fl/fl^ and LysM;Rgs12^fl/fl^ mice.

Manuscript revisions:

- Experiments completed and data incorporated into Figure 1 as panels N-P.

- Data used to update Results section.

- Discussion updated.

- New methodology added to Methods section.

C) Assessing the effects of Rgs12 deletion or overexpression on osteoclastic bone resorption

Findings: Rgs12-deleted osteoclast precursors cultured ex vivo showed significantly reduced bone resorption activity whereas Rgs12 overexpressed in RAW264.7 cells increased bone resortion.

Manuscript revisions:

- Experiments completed and incorporated into Figure 2 as panels E-F and K-L.

- Data used to update Results section.

- Discussion updated.

- New methodology added to Methods section.

2) What is the effect of 1 to 5 days of RANKL treatment on levels of Rgs12 mRNA and protein in BMM cultures compared to cultures treated with M-CSF alone? This is a fundamental experiment that the authors do not present. Are levels increased, unchanged or decreased by RANKL? The authors seem to be arguing that RANKL increases Rgs12 levels since overexpression of this protein enhanced and deletion decreased RANKL-induced osteoclastogenesis. However, the authors really need to show this kind of data to confirm this hypothesis.

To address this gap, we measured Rgs12 expression during osteoclast differentiation. BMMs obtained from either LysM;Rgs12^+/+^ or Rgs12^fl/fl^ mice was differentiated using M-CSF and RANKL for up to 5 days. Rgs12 transcript and protein levels at different time-points of differentiation were measured by qPCR and western blotting, respectively.

Findings: Rgs12 expression as measured by western blotting and qPCR was increased over the course of osteoclast differentiation.

Manuscript Revisions:

- Experiments completed and data incorporated into Figure 2 as panels A-B.

- Data used to update Results section.

References:

1) Gonzales, F. and Karnovsky, M. J. Electron microscopy of osteoclasts in healing fracturees of rat bone. J Biophys Biochem Cytol 9, 299-316 (1961).

2) An, E., Narayanan, M., Manes, N. P. and Nita-Lazar, A. Characterization of functional reprogramming during osteoclast development using quantitative proteomics and mRNA profiling. Mol Cell Proteomics 13, 2687-2704, doi:10.1074/mcp.M113.034371 (2014).

3) Ng, A. Y. et al. Comparative Characterization of Osteoclasts Derived From Murine Bone Marrow Macrophages and RAW 264.7 Cells Using Quantitative Proteomics. JBMR Plus 2, 328-340, doi:10.1002/jbm4.10058 (2018).

4) Lemma, S. et al. Energy metabolism in osteoclast formation and activity. Int J Biochem Cell Biol 79, 168-180, doi:10.1016/j.biocel.2016.08.034 (2016).

5) Ishii, K. A. et al. Coordination of PGC-1beta and iron uptake in mitochondrial biogenesis and osteoclast activation. Nat Med 15, 259-266, doi:10.1038/nm.1910 (2009).

6) Bartell, S. M. et al. FoxO proteins restrain osteoclastogenesis and bone resorption by attenuating H2O2 accumulation. Nat Commun 5, 3773, doi:10.1038/ncomms4773 (2014).

7) Elefteriou, F. and Yang, X. Genetic mouse models for bone studies--strengths and limitations. Bone 49, 1242-1254, doi:10.1016/j.bone.2011.08.021 (2011).

8) Huang, W. and Olsen, B. R. Skeletal defects in Osterix-Cre transgenic mice. Transgenic Res 24, 167-172, doi:10.1007/s11248-014-9828-6 (2015).

9) Ohishi, M. et al. Suppressors of cytokine signaling-1 and -3 regulate osteoclastogenesis in the presence of inflammatory cytokines. J Immunol 174, 3024-3031 (2005).

10) Albers, J. et al. Canonical Wnt signaling inhibits osteoclastogenesis independent of osteoprotegerin. J Cell Biol 200, 537-549, doi:10.1083/jcb.201207142 (2013).

11) Qi, B. et al. Ablation of Tak1 in osteoclast progenitor leads to defects in skeletal growth and bone remodeling in mice. Sci Rep 4, 7158, doi:10.1038/srep07158 (2014).

12) Cong, Q. et al. p38alpha MAPK regulates proliferation and differentiation of osteoclast progenitors and bone remodeling in an aging-dependent manner. Sci Rep 7, 45964, doi:10.1038/srep45964 (2017).

13) Matsumoto, Y. et al. RANKL coordinates multiple osteoclastogenic pathways by regulating expression of ubiquitin ligase RNF146. J Clin Invest 127, 1303-1315, doi:10.1172/JCI90527 (2017).

14) Rohatgi, N. et al. ASXL1 impairs osteoclast formation by epigenetic regulation of NFATc1. Blood Adv 2, 2467-2477, doi:10.1182/bloodadvances.2018018309 (2018).

15) Sun, Y. X. et al. Deletion of Nrf2 reduces skeletal mechanical properties and decreases load-driven bone formation. Bone 74, 1-9, doi:10.1016/j.bone.2014.12.066 (2015).

16) Ibanez, L. et al. Effects of Nrf2 deficiency on bone microarchitecture in an experimental model of osteoporosis. Oxid Med Cell Longev 2014, 726590, doi:10.1155/2014/726590 (2014).

17) Kim, J. H., Singhal, V., Biswal, S., Thimmulappa, R. K. and DiGirolamo, D. J. Nrf2 is required for normal postnatal bone acquisition in mice. Bone Res 2, 14033, doi:10.1038/boneres.2014.33 (2014).

18) Rana, T., Schultz, M. A., Freeman, M. L. and Biswas, S. Loss of Nrf2 accelerates ionizing radiation-induced bone loss by upregulating RANKL. Free Radic Biol Med 53, 2298-2307, doi:10.1016/j.freeradbiomed.2012.10.536 (2012).

19) Kanzaki, H. et al. Nuclear Nrf2 induction by protein transduction attenuates osteoclastogenesis. Free Radic Biol Med 77, 239-248, doi:10.1016/j.freeradbiomed.2014.09.006 (2014).

20) Kanzaki, H., Shinohara, F., Kajiya, M. and Kodama, T. The Keap1/Nrf2 protein axis plays a role in osteoclast differentiation by regulating intracellular reactive oxygen species signaling. J Biol Chem 288, 23009-23020, doi:10.1074/jbc.M113.478545 (2013).

21) Hyeon, S., Lee, H., Yang, Y. and Jeong, W. Nrf2 deficiency induces oxidative stress and promotes RANKL-induced osteoclast differentiation. Free Radic Biol Med 65, 789-799, doi:10.1016/j.freeradbiomed.2013.08.005 (2013).

22) Abiko, Y., Miura, T., Phuc, B. H., Shinkai, Y. and Kumagai, Y. Participation of covalent modification of Keap1 in the activation of Nrf2 by tert-butylbenzoquinone, an electrophilic metabolite of butylated hydroxyanisole. Toxicol Appl Pharmacol 255, 32-39, doi:10.1016/j.taap.2011.05.013 (2011).

23) Simicevic, J. and Deplancke, B. Transcription factor proteomics-Tools, applications, and challenges. Proteomics 17, doi:10.1002/pmic.201600317 (2017).

24) Rushworth, G. F. and Megson, I. L. Existing and potential therapeutic uses for N-acetylcysteine: the need for conversion to intracellular glutathione for antioxidant benefits. Pharmacol Ther 141, 150-159, doi:10.1016/j.pharmthera.2013.09.006 (2014).

25) Lee, N. K. et al. A crucial role for reactive oxygen species in RANKL-induced osteoclast differentiation. Blood 106, 852-859, doi:10.1182/blood-2004-09-3662 (2005).

26) Tai, T. W. et al. Reactive oxygen species are required for zoledronic acid-induced apoptosis in osteoclast precursors and mature osteoclast-like cells. Sci Rep 7, 44245, doi:10.1038/srep44245 (2017).

27) Cao, J. J. and Picklo, M. J. N-acetylcysteine supplementation decreases osteoclast differentiation and increases bone mass in mice fed a high-fat diet. J Nutr 144, 289-296, doi:10.3945/jn.113.185397 (2014).

28) Kim, H. R., Kim, K. W., Kim, B. M., Lee, K. A. and Lee, S. H. N-acetyl-l-cysteine controls osteoclastogenesis through regulating Th17 differentiation and RANKL in rheumatoid arthritis. Korean J Intern Med, doi:10.3904/kjim.2016.329 (2017).

29) Gleixner, A. M. et al. N-Acetyl-l-Cysteine Protects Astrocytes against Proteotoxicity without Recourse to Glutathione. Mol Pharmacol 92, 564-575, doi:10.1124/mol.117.109926 (2017).

30) Halasi, M. et al. ROS inhibitor N-acetyl-L-cysteine antagonizes the activity of proteasome inhibitors. Biochem J 454, 201-208, doi:10.1042/BJ20130282 (2013).

[Editors' note: further revisions were requested prior to acceptance, as described below.]1) Figure 4A western is not at all convincing that deletion of Rgs12 increases Nrf2 activity and this is a major premise of the paper – a critical figure; we would like to see the difference more clearly

A better quality western blot was substituted into Figure 4A.

2) The authors did NOT respond to the issue of the proper control for the LysMCre Rgs12 targeted deletion by skeletal phenotyping; the appropriate control is the LysMCreRgs12+/+, that should be relatively easy to phenotype and exclude a transgenic Cre effect. All that you say is that the animals are not different in size. It should be relatively straightforward since these are the controls to have bone density data.

To exclude a possible transgenic Cre effect, we characterized and compared the bone histomorphometry of *LyzM^Cre^;Rgs12^+/+^* and *Rgs12^flox/flox^* mice, and found no statistical difference in the quantitative micro-CT measurements. Therefore, either of these mouse genotypes would be suitable as negative controls in this study. This finding was updated in the manuscript (subsection “Targeted deletion of Rgs12 selectively reduced osteoclast formation and increased trabecular bone mass”, first paragraph).

Data added as Figure 1—figure supplement 1. Corresponding figure legend added to manuscript.

Data added to Figure 1—source data.